# Global influence of soil texture on ecosystem water limitation

Fabian J. P. Wankmüller[1], Louis Delval[2], Peter Lehmann[1], Martin J. Baur[3,4], Andrea Cecere[2], Sebastian Wolf[1], Dani Or[1,5], Mathieu Javaux[2,6 ✉] & Andrea Carminati[1 ✉]

Low soil moisture and high vapour pressure deficit (VPD) cause plant water stress and lead to a variety of drought responses, including a reduction in transpiration and photosynthesis[1,2]. When soils dry below critical soil moisture thresholds, ecosystems transition from energy to water limitation as stomata close to alleviate water stress[3,4]. However, the mechanisms behind these thresholds remain poorly defined at the ecosystem scale. Here, by analysing observations of critical soil moisture thresholds globally, we show the prominent role of soil texture in modulating the onset of ecosystem water limitation through the soil hydraulic conductivity curve, whose steepness increases with sand fraction. This clarifies how ecosystem sensitivity to VPD versus soil moisture is shaped by soil texture, with ecosystems in sandy soils being relatively more sensitive to soil drying, whereas ecosystems in clayey soils are relatively more sensitive to VPD. For the same reason, plants in sandy soils have limited potential to adjust to water limitations, which has an impact on how climate change affects terrestrial ecosystems. In summary, although vegetation–atmosphere exchanges are driven by atmospheric conditions and mediated by plant adjustments, their fate is ultimately dependent on the soil.

Terrestrial ecosystems are home to most species on Earth[5], have a key role in the global climate system[6] and provide annual ecosystem services estimated to be approximately equivalent to the annual global gross domestic product[7]. However, due to climate change, land ecosystems are experiencing higher temperature increases than the global (land–ocean) average[6], contributing to widespread increases in vapour pressure deficit (VPD)[8,9] and drought frequency[10–12], which lead to amplified water limitation[13], reduced global vegetation growth[9], land degradation and food insecurity in many regions[6]. Low soil moisture (volumetric water content, $\theta$) and high VPD are considered the two main drivers of plant water stress, but their relative importance in ecosystem water limitation is debated[1,2,14,15]. While the exchange of water between vegetation and the atmosphere is initially driven by energy availability, soil drying below a critical soil moisture threshold ($\theta_{crit}$) limits the soil–plant water supply, which causes stomata to downregulate transpiration ($T$). The decrease in transpiration is accompanied by reduced gross primary production (GPP) and reduced evaporative cooling, resulting in a feedback between ecosystem water limitation and climate warming[3,16,17]. Therefore, critical soil moisture thresholds have a crucial role in the vegetation and climate of terrestrial ecosystems. However, the key mechanisms controlling these thresholds remain poorly defined at the ecosystem scale.

The closure of stomata at critical soil moisture thresholds, or at the corresponding critical soil water potential thresholds ($\psi_{crit}$), is triggered by a decrease in leaf water potential ($\psi_{leaf}$) and soil–plant hydraulic conductance ($K_{soil+plant}$)[18–20]. The decrease in $\psi_{leaf}$ depends on the soil water potential ($\psi_{soil}$), the upstream hydraulic conductances (soil and plant) and the actual transpiration rate. Critical soil water thresholds ($\theta_{crit}$ and $\psi_{crit}$) are therefore influenced by atmospheric, plant and soil variables. Relevant variables include the: (1) atmospheric conditions driving the transpiration stream (that is, solar radiation, VPD, boundary layer thickness and conductance); (2) plant traits mediating the transpiration rate (that is, physiological and hydraulic traits, such as stomatal sensitivity to $\psi_{leaf}$, and root–shoot investment); and (3) soil hydraulic properties supplying it (that is, soil water retention and hydraulic conductivity curves). As water flows from the soil into the roots and along the xylem to the leaves and stomatal cavities through resistances in series (Fig. 1a), the element with the lowest hydraulic conductance determines the total conductance of the soil–plant system. The conductance of each element decreases with the respective element water potential, and a sharp drop in water potential occurs across the limiting hydraulic element. This drop can lead to a further reduction in downstream hydraulic conductance, and eventually to low leaf water potentials, triggering the downregulation of fluxes due to the decrease in canopy conductance ($g_c$) caused by stomatal closure.

The relative importance of the conductance of each element (that is, $K_{soil}$, $K_{root}$, $K_{stem}$ and $K_{leaf}$) not only affects the onset of water limitation, but also shapes the relative importance of VPD versus soil moisture limitation. When the limiting hydraulic conductance resides within the plant (in the roots or shoot), the leaf water potential is primarily affected by VPD, which determines the water flow rate, because the large dissipation in water potential occurs within the plant and is proportional

[1]Institute of Terrestrial Ecosystems, ETH Zurich, Zurich, Switzerland. [2]Earth and Life Institute, Environmental Sciences, UCLouvain, Ottignies-Louvain-la-Neuve, Belgium. [3]Department of Geography, University of Cambridge, Cambridge, UK. [4]Conservation Research Institute, University of Cambridge, Cambridge, UK. [5]Department of Civil and Environmental Engineering, University of Nevada, Reno, NV, USA. [6]Agrosphere IBG-3, Forschungszentrum Jülich, Jülich, Germany. ✉e-mail: mathieu.javaux@uclouvain.be; andrea.carminati@usys.ethz.ch

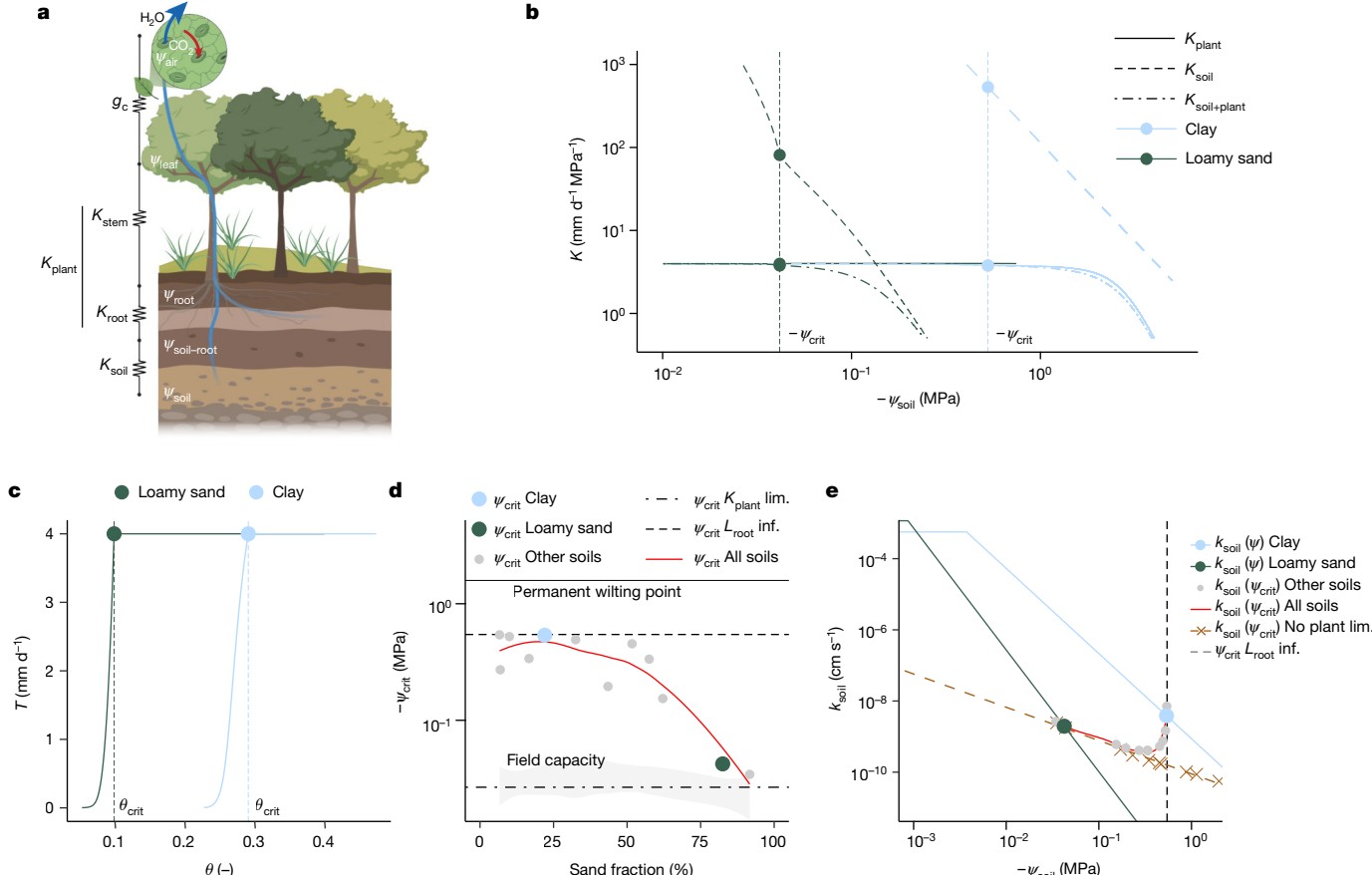

**Fig. 1 | The relative importance of soil and plant hydraulics in ecosystem water limitation varies with soil texture. a**, Critical thresholds of ecosystem water limitation depend on the relative importance of soil ($K_{soil}$) and plant ($K_{plant}$: $1/K_{plant} = 1/K_{root} + 1/K_{stem}$) hydraulic conductance in triggering a decrease in canopy conductance ($g_c$). **b**, The relative importance of $K_{soil}$ and $K_{plant}$ depends on soil texture. The steeper decline of $K_{soil}$ in coarse soils (that is, loamy sands) triggers ecosystem water limitation at less negative critical soil water potential thresholds ($\psi_{crit}$) than in fine soils (that is, clays), also translating into differences in critical soil moisture thresholds ($\theta_{crit}$). **c**, $\theta_{crit}$ is defined as the minimum soil moisture ($\theta$) at which the soil–plant hydraulic system can supply water at the rate of the potential transpiration rate (that is, 4 mm d$^{-1}$). Owing to different soil hydraulic properties, fine soils show ecosystem water limitation at higher $\theta$ than coarse soils. **d**,**e**, $\psi_{crit}$ (**d**) and the soil hydraulic conductivity at $\psi_{crit}$, $k_{soil}$ ($\psi_{crit}$), (**e**) as a function of soil texture. Neglecting soil water limitations (equivalent to

$K_{soil} \gg K_{plant}$), either by assuming an infinite root length ($L_{root}$ inf., black dashed line) or an early limitation by plant hydraulic conductance ($K_{plant}$ lim., black dashed–dotted line), we would expect a uniform $\psi_{crit}$, independent of soil texture, close to the permanent wilting point (solid black line), and field capacity (light grey polygon), respectively. Considering soil and plant water limitations (default simulation), we would expect $\psi_{crit}$ to become less negative with an increasing sand fraction ($\psi_{crit}$ all soils, red curve representing a local polynomial regression fitting) as a result of the contrasting soil hydraulic conductivity curves. **e**, In the coarsest-textured soils, $k_{soil}$ and $\psi_{crit}$ follow a linear decline (dashed brown line) corresponding to simulations excluding any plant hydraulic limitations (no plant lim.). In very fine-textured soils, $\psi_{crit}$ converges to a constant value controlled by the water potential at which plants lose conductivity (black dashed vertical line, as in **d**, representing $L_{root}$ inf.). Illustration in **a** created using BioRender (https://biorender.com).

to the transpiration rate. In this case, the onset of water limitation is expected to be particularly sensitive to VPD and less to soil drying. Instead, when the soil is the limiting element responsible for triggering stomatal closure, transpiration is particularly sensitive to soil drying and, in addition, critical soil water thresholds would be affected by the soil hydraulic conductivity and its dependence on the water potential.

Currently, it is under debate which element of the soil–plant continuum is the hydraulic limit triggering stomatal closure. Some studies have highlighted the role of plant tissues—that is, the leaves[21,22], the xylem[23] or the roots[24]—while others have pointed to the limiting role of the soil[25,26] and soil–root interface[27]. This ambiguity regarding the limiting hydraulic element of the soil–plant continuum is mechanistically linked to the debate about the relative importance of soil drying versus VPD for the onset of ecosystem water limitation. Note that the ranking of which soil–plant element is the hydraulic limit varies with time (for example, there is no soil limitation when the soil is wet, but $K_{soil}$ may become limiting as the soil dries below the critical soil water thresholds during the growing season).

The key principles of soil–plant hydraulics and water use regulation are well established at the plant scale, and have been successfully applied in irrigation management[28] and implemented in models of soil–plant water relations[20,29–32]. However, extending these principles to natural systems at the ecosystem scale remains challenging. This is primarily due to the large uncertainty in the key hydraulic variables operating at this scale that obscure the dominant mechanism behind critical soil water thresholds. In particular, uncertainties in soil and plant hydraulic properties, such as soil hydraulic conductivity functions and root length density, soil spatial heterogeneity and complex species composition, pose challenges to unambiguously identifying the limiting hydraulic element along the soil–plant–atmosphere continuum. Notwithstanding these uncertainties, the strong dependence of the soil hydraulic conductivity curve on soil texture allows testing of the limiting role of the soil in natural ecosystems by investigating the relationship between critical soil water thresholds and soil texture. Recent developments in terrestrial monitoring networks, such as eddy covariance and sap flow measurements, and remote sensing

have enabled the identification of $\theta_{crit}$ across soil textures, climates and biomes[4,33,34], providing an unprecedented opportunity for testing these variables and their dependence on soil texture at the ecosystem scale.

Here, we hypothesize that the steep decline in soil hydraulic conductivity with soil water potential (Fig. 1b,e) triggers the downregulation of water fluxes—that is, of transpiration rate and root water uptake. This is particularly relevant in coarse-textured soils, whose hydraulic conductivity curves decline particularly steeply with decreasing water potential, whereas it is less important in fine-textured soils, in which it is rather the decline in plant hydraulic conductance ($K_{plant}$) that limits plant water use (Fig. 1b). The hydraulic conductances ($K_{soil}$ and $K_{plant}$) are defined as the ratio between the transpiration rate and the difference in water potential across the respective element (soil and plant), and the total conductance of the soil–plant continuum ($K_{soil+plant}$) is the harmonic mean of the two conductances. The water potential across soil and plant is calculated by solving the flow equation in both compartments for a given potential transpiration rate and includes the notion that the hydraulic conductivity in soil and plants declines with declining water potential[25]. Transpiration is limited when the $K_{soil}$ or $K_{plant}$ decline sufficiently enough to impact the relationship between $T$ and $\psi_{leaf}$ (dotted line in Fig. 1b, corresponding to a given soil water potential, $\psi_{crit}$). Note that the limiting hydraulic element is not required to have the lowest absolute conductance among all hydraulic elements to be able to induce an effective loss in the overall $K_{soil+plant}$ (ref. 25). This can be seen even in coarse-textured soils (Fig. 1b, loamy sand), where the $K_{soil}$ at $\psi_{crit}$ is still higher than $K_{plant}$. However, if the stomata were not closing, the $K_{soil}$ would drop almost vertically because the soil could no longer sustain the transpiration demand (note the marked decline in $K_{soil}$ right before $\psi_{crit}$). Therefore, the steep loss in $K_{soil}$ is enough to initiate an initial decline in $K_{soil+plant}$ and to trigger stomatal closure even when $K_{soil} > K_{plant}$. In other words, stomatal closure prevents an excessive drop in soil conductance which could have much worse consequences for plant water use and functioning[25,27,35].

A consequence of this analysis is that the relative importance of soil versus plant hydraulic limitation is expected to be soil-texture-dependent. Specifically, we hypothesize that $\theta_{crit}$ (Fig. 1c) and $\psi_{crit}$ (Fig. 1b,d,e) are functions of soil texture, with $\psi_{crit}$ becoming less negative with increasing sand fraction (Fig. 1d, red line). Note that, due to the texture-dependent relationship between $\theta$ and $\psi$, the lower $\theta_{crit}$ values in coarse-textured soils correspond to less negative $\psi_{crit}$ values compared to fine-textured soils. By contrast, if stomatal closure was triggered by an early decline in $K_{plant}$, or if soil hydraulic limitation was negligible (thanks to an infinitely large root surface), we would expect $\psi_{crit}$ to be uniform across soil textures. More precisely, $\psi_{crit}$ would be close to the field capacity in the first case (dashed–dotted line in Fig. 1d) and close to the permanent wilting point in the second case (dashed line). The filled circles in Fig. 1d are model calculations for each soil textural class, while the red line is an interpolation. The larger blue and green circles correspond to the $\psi_{crit}$ of clay and loamy sand, as calculated in Fig. 1b.

The relative importance of soil and plant hydraulic limitations and the specific role of soil texture is well represented in the relationship between $\psi_{crit}$ and the soil hydraulic conductivity. Figure 1e shows the $\psi_{crit}$ and $k(\psi_{crit})$ of each soil texture (closed circles, corresponding to the circles in Fig. 1d). In the coarsest-textured soils, $\psi_{crit}$ is close to the field capacity. With finer soil texture, $\psi_{crit}$ becomes more negative and $k_{soil}(\psi_{crit})$ decreases, as the conductivity curves become flatter. In very fine soils, $\psi_{crit}$ converges to a constant value (dashed grey line) controlled by the water potential at which plants lose conductivity, causing $k_{soil}(\psi_{crit})$ to increase again. Overall, the trajectory of this relationship (solid red line) is constrained by two lines, one (dashed brown) indicating soil hydraulic constraints alone—that is, in the absence of any plant hydraulic limitations—and one vertical line (dashed grey) determined by plant hydraulic limitations. This is a key figure because: (1) it provides a mechanistic explanation of soil water limitation; and (2) it shows the relative importance of soil and plant hydraulics in a new

and clear way, suggesting that transpiration tends to be soil limited in coarse-textured soils and plant limited in fine-textured soils. Note that Fig. 1b–e result from a model[25] that hypothesizes that transpiration is limited by a decline in the conductance of either soil or plants (Extended Data Fig. 1). The model thus predicts that soil water thresholds and the relative importance of soil and plants are soil-texture specific.

To test the hypothesized soil texture dependence of $\psi_{crit}$, we combined global observations of $\theta_{crit}$, obtained from two complementary measurements of (evapo)transpiration, with soil–plant hydraulic modelling. We analysed whether the expected dependence of $\psi_{crit}$ and $\theta_{crit}$ (Fig. 1b–e) is visible at the tree and ecosystem scale across biomes and climates, where plant selection and adaptation to soil and climate may mask the effects of soil texture. In addition to demonstrating the effect of soil texture on critical soil water thresholds[4,34], our study aimed to determine the mechanistic basis of these thresholds, including understanding the effects of plant trait plasticity and future climate on ecosystem water limitation. As a first step, we simulated critical soil water thresholds by varying the soil hydraulic properties alone (Supplementary Table 2). Next, we tested the effect of plant trait adjustments (root length density and plant vulnerability) on critical soil water thresholds. Afterwards, we analysed the relative importance of VPD versus soil moisture across soil textures. Finally, we assessed the impact of future climate on ecosystem water limitation by mapping the expected changes in $\theta_{crit}$ in response to future VPD conditions and evaluated the implications for vegetation and ecosystem fluxes.

## Soil texture modulates soil water thresholds

In line with previous studies[4,34,36], our results show that critical soil moisture thresholds across the globe are strongly dependent on soil texture. In both datasets (hereafter referred to as FLUXNET (FN) and SAPFLUXNET (SFN)), $\theta_{crit}$ is inversely related to the gravimetric sand fraction (%) and decreases from more than 0.2 for clayey soils to less than 0.1 in sandy soils (Fig. 2a). Clay soils aside (SFN), the simulated soil-texture-specific estimates of $\theta_{crit}$ were in good agreement with the observed $\theta_{crit}$ (Fig. 2b, see figure caption for details on linear regressions). Because all model parameters were kept constant, aside from the soil hydraulic properties that varied with the local soil texture at each site, the strong soil texture dependence of $\theta_{crit}$ indicates a prominent role of soil hydraulic properties in ecosystem water limitation.

The variability in $\theta_{crit}$ varied with soil texture, being inversely related to the sand fraction. The soil texture dependence of $\theta_{crit}$-variability (span of $S_{\theta crit}$ in Fig. 2c) is explained by the sensitivity of $\theta_{crit}$ to plant hydraulic variability (plant vulnerability, $\psi_{x*}$, and active root length, $L_{root}$) and soil hydraulic properties, which are expected to vary in each soil textural class (for example, coarse versus fine sand). Varying $\psi_{x*}$ (from −1.5 to −5 MPa) affected $\theta_{crit}$ such that $S_{\theta crit}$ was larger in fine (approximately 0.15) than in coarse (less than 0.05) textured soils (Fig. 2c), while changing $L_{root}$ affected $\theta_{crit}$ without exhibiting a soil texture dependence. Variations in $\psi_{x*}$ have a stronger effect in fine-textured soils because ecosystems in these soils are expected to be more plant limited. This analysis shows that the impact of plant hydraulic adjustment on $\theta_{crit}$ depends on soil texture. While $\theta_{crit}$ could be considerably shaped by plant vulnerability adjustments in fine-textured soils, the effect in coarse-textured soils tends to be marginal. Moreover, the predicted $S_{\theta crit}$, considering the variability in soil hydraulic parameters in each textural class (more than 0.2 in clay to less than 0.1 in sand) agrees well with the observed $\theta_{crit}$ variability (Fig. 2c). This demonstrates how sensitive soil water thresholds are to soil hydraulic properties, and calls for direct measurements of soil hydraulic properties, which are an essential element in predicting site-specific critical soil water thresholds.

In addition to $\theta_{crit}$, $\psi_{crit}$ also depends on soil texture and exhibits the hypothesized trend (Fig. 1d). The median of $\psi_{crit}$ ranged from values typically associated with field capacity in sandy soils (about −0.03 MPa) to values approaching the permanent wilting point in clay soils

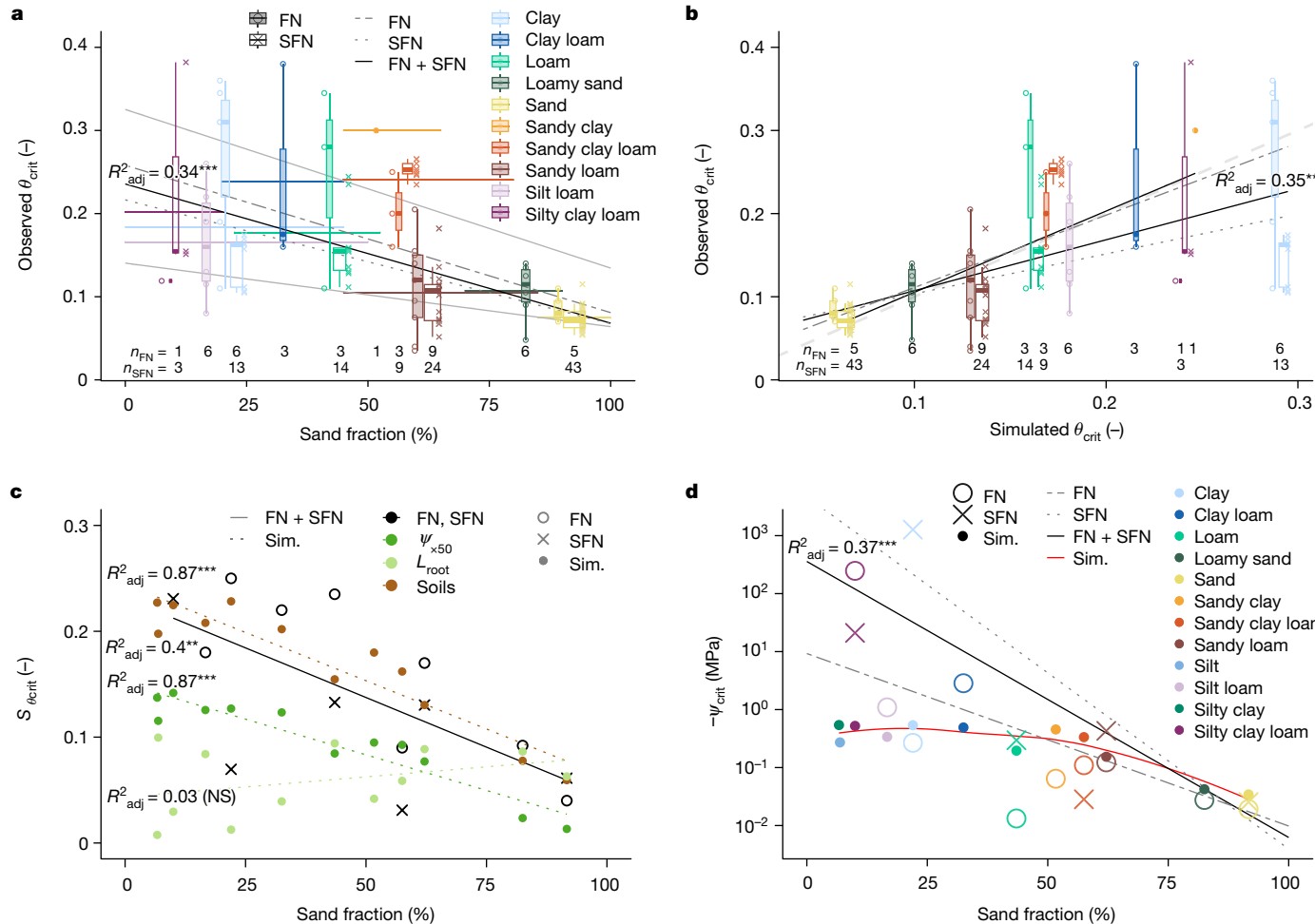

**Fig. 2 | Soil hydraulic conductivity controls critical soil water thresholds of ecosystem water limitation. a**, Global critical soil moisture thresholds correlated with the mean (range indicated by horizontal error bars) gravimetric sand fraction per soil textural class (linear regression of $\theta_{crit}$ to mean sand fraction for all observations—that is, FN + SFN—for soil texture specific maximum and minimum $\theta_{crit}$, and for all FN and SFN $\theta_{crit}$ separately, are shown as solid black, solid grey, dashed–dotted grey and dotted grey lines, respectively). **b**, Critical soil moisture thresholds are well predicted by applying the model to the site-specific soil textural information (linear regressions of observed-to-simulated $\theta_{crit}$, considering all soil textures, and by excluding clay soils, are shown as solid black, while linear regressions of FN and SFN separately are shown as dashed–dotted and dotted grey lines, respectively). **c**, The sand fraction dictates the span of observed (FN, SFN) and simulated (sim.) critical soil moisture thresholds ($S_{\theta crit}$) and the difference between maximum and minimum $\theta_{crit}$ per soil textural class. $S_{\theta crit}$ is a measure of the $\theta_{crit}$ variability of the observations (black) and of the $\theta_{crit}$ sensitivity to the model parameters, such as varying soil hydraulic properties in a soil textural class (brown), and varying plant traits (plant vulnerability, $\psi_{x50}$, and root length, $L_{root}$, in green). The negative slopes of the linear regressions of $S_{\theta crit}$ (except for $L_{root}$)

demonstrate the decreasing variability and sensitivity of $\theta_{crit}$ with sand fraction, in both observation and simulation (solid black and coloured dotted lines, respectively). **d**, The observed median critical soil water potential thresholds ($\psi_{crit}$) for FN (open circles) and SFN (crosses) confirm the expected decline with increasing sand fraction (note the simulations (filled circles and red solid curve) correspond to Fig. 1d, and the $\log_{10}$-transformed linear regressions for all $\psi_{crit}$, not only the medians, are shown as solid black, with FN and SFN shown separately as dashed–dotted and dotted grey lines, respectively). **a**,**b**, The data are presented as grouped (FN, SFN) box plots (the thick solid line represents the median, the lower and upper hinges correspond to the first and third quartiles, with the whiskers extending to the highest or lowest value, respectively, but no further than 1.5 times the interquartile range, and the width of the boxes scales with the square root of the number of observations in each soil textural class) in combination with individual observations displayed as points along the boxes ($n_{FN}$ and $n_{SFN}$ indicate the number of FN and SFN $\theta_{crit}$ observations per soil textural class, respectively, while the number of sites per soil textural class is given in Supplementary Table 2). **a**–**d**, Adjusted $R^2$ values ($R^2_{adj}$) and two-sided $P$ values of the linear regression slopes (regression $t$-test) are indicated. *$P < 0.05$, **$P < 0.01$, ***$P < 0.001$. NS, not significant ($P > 0.05$).

(about −0.6 MPa) (Fig. 2d). $\psi_{crit}$ is inversely related to the sand fraction (note the logarithmic scale), meaning that ecosystems in sandy soils become water limited at less negative soil water potentials than in fine-textured soils. This is explained by the steeper decline in soil hydraulic conductivity in coarse soils, as shown in Fig. 1e.

Notably, the two monitoring networks, despite their differences in methodology and scale, provided consistent results and confirmed the hypothesis on the texture-dependent relative importance of soil and plant hydraulics in controlling the onset of ecosystem water limitation. Evident exceptions were the estimates of $\theta_{crit}$ and $\psi_{crit}$ in clay, which were particularly low (that is, dry) in the SFN data. Unlike FN, which

reports evapotranspiration, SFN measurements only determine water fluxes through plants—more precisely, trees. In this case, considering only the volumetric water content of the topmost soil layer might not have been representative enough of the soil water limitations in clay. In other words, root water uptake from deeper soil layers might have been effective enough to supply transpiration at the maximum rate. Note that the assumption of our analysis is that the initial decline in transpiration is driven by drying of the topsoil, where the root density is typically highest. Water uptake from deeper soil layers is important in the rate of decline in transpiration and in plant stress, but this was not investigated here.

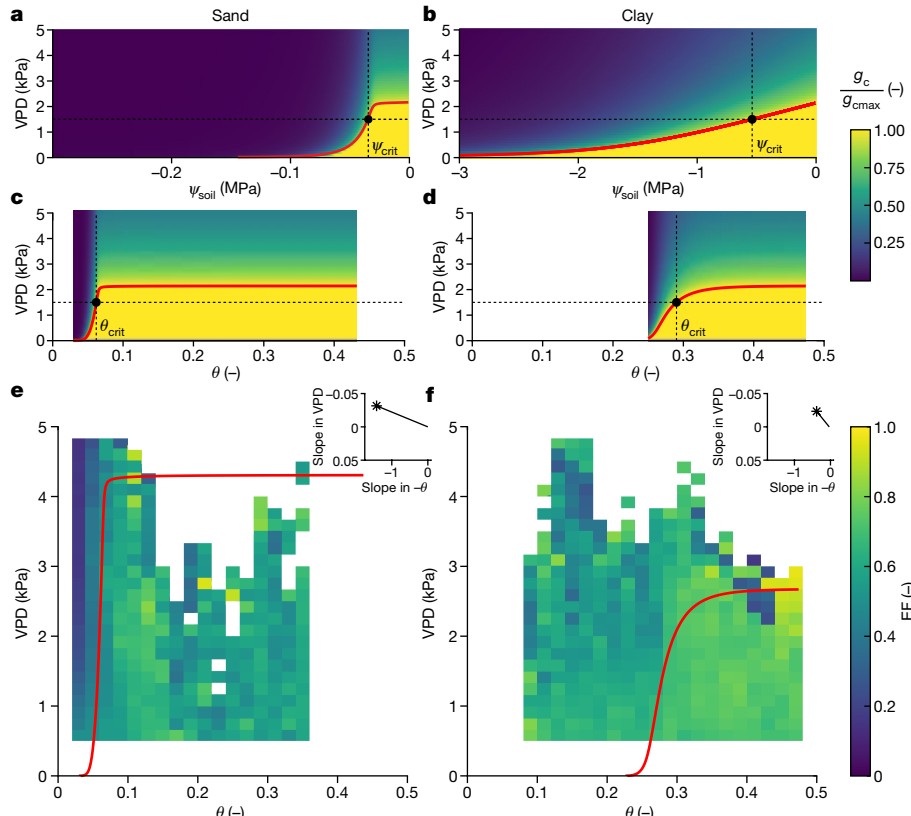

**Fig. 3 | Soil texture shapes the relative importance of VPD and soil moisture.** **a**–**d**, Soil texture drives the relative importance of VPD versus the soil moisture ($\theta$) limitation of ecosystem fluxes (that is, the downregulation of $g_c$ from its maximum $g_{cmax}$) and involves implications of future climate on terrestrial ecosystems (Fig. 4). Ecosystems in fine-textured soils (that is, clays (**b**,**d**)) are expected to be comparatively more sensitive to VPD than those in coarse soils (that is, sands (**a**,**c**)), while ecosystems in coarse-textured soils are expected to be comparatively more sensitive to soil drying than in fine soils because critical soil water potentials ($\psi_{soil}$) are more negative (note the 10-fold different $x$-axis limit in **a**,**b**), and the $g_c$ downregulation is more gradual (softer colour transitions), in fine- than in coarse-textured soils. **e**,**f**, The evaporative fraction (EF) from eddy covariance data shows different responses to the two environmental drivers, $\theta$ and VPD, for the two contrasting soil textures (sand (**e**) and clay (**f**) sites, median of five FN sites). The evaporative fraction declines in both soil textures within a narrow range of soil moisture, but more sharply and at lower absolute water contents in the sand sites than in the clay sites. The simulations of transpiration rate as a function of VPD and $\theta$ (red line) agree well with the observed decline in the evaporative fraction around $\theta_{crit}$. The inset plots show the median relative sensitivity of evaporative fraction to VPD and $\theta$, confirming the stronger relative contribution of soil hydraulic limitation in coarse-textured compared to fine-textured soils.

## Relative importance of VPD and soil moisture

Our analysis of critical soil water thresholds reveals the important role of soil texture through the emergent control of soil hydraulic conductivity on the onset of ecosystem water limitation. This offers new insights into ecosystem responses to drought, such as the sensitivity of ecosystems to increasing atmospheric water demand (that is, VPD) and to more frequent soil drying. Ecosystems in coarse-textured soils are expected to be more sensitive to soil drying and less to VPD in comparison to ecosystems in fine-textured soils (Fig. 3a–d). For the same reason, they are less sensitive to plant internal hydraulic adjustments (Fig. 2c). In coarse-textured soils, a small change in water content under dry conditions results in a large decrease in water potential and a large decrease in soil hydraulic conductivity. This results in a marked drop in transpiration as a function of both soil moisture and soil water potential and a smaller sensitivity to VPD (Fig. 3a,c). The sensitivity to soil drying is less in fine-textured soils, which have less-steep hydraulic conductivity curves (Fig. 1e) and show a more gradual decline in transpiration as a function of soil water content and soil water potential (Fig. 3b,d). Compared to coarse-textured soils, ecosystems in fine-textured soils are therefore relatively more sensitive to VPD and less to soil drying.

Based on this mechanism, we expected an impact of soil texture on the relative importance of VPD versus soil moisture limitation.

We analysed the evaporative fraction of ecosystems in contrasting soil textures and compared their VPD versus soil moisture limitations. The predicted relationship between transpiration, soil moisture and VPD agreed well with the observed (Fig. 3e,f). The simulated onset of hydraulic limitation (red line) demarks well the observed transition between energy-driven (lower right corner, yellowish green) and water-limited (upper left corner, blueish) fluxes in both soil textures. The observations clearly show that the evaporative fraction is comparatively more sensitive to soil drying in sandy soils because the evaporative fraction decreases sharply with soil moisture, and the relative sensitivity of evaporative fraction with respect to $\theta$ is greater than with respect to VPD. In clay, the evaporative fraction decreases comparatively more gradually with soil moisture, and the slope of the evaporative fraction with respect to $\theta$ is comparatively more similar to the slope with respect to VPD. The good agreement between the simulations and the different sensitivities of $\theta_{crit}$ to VPD and soil moisture observed in the coarse- and fine-textured soils suggest an important control of soil texture on VPD versus soil moisture limitation. Our analysis therefore contributes a new perspective to the ongoing debate on whether ecosystems are water limited by soil moisture or VPD[14,15]. It also suggests that ecosystems in fine-textured soils are likely more affected by increases in VPD than ecosystems in coarse-textured soils. Hence, soil-texture-specific soil hydraulic properties should be considered when investigating the impacts of climate change on terrestrial ecosystems.

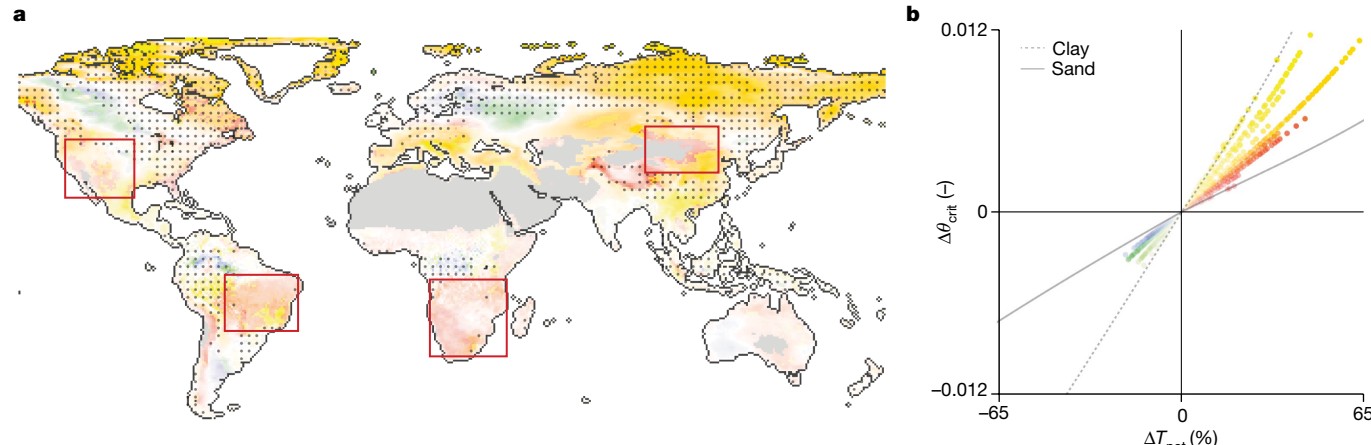

**Fig. 4 | The global sensitivity of critical soil moisture thresholds to climate change depends on soil texture. a**, Predicted changes in global critical soil moisture thresholds ($\Delta\theta_{crit}$) in response to changes in VPD from current (2005–2014) to future (2060–2069) climate (SSP2-4.5 scenario). The four rectangles highlight regions where we expect the highest amplification of ecosystem vulnerability to drought due to increasing VPD. These regions will experience an increase in atmospheric drying, but show limited buffer capacity (small $\Delta\theta_{crit}$) due to the coarseness of their soil texture. Hyperarid deserts (dark grey, aridity index (AI) ≤ 0.05) were excluded. In humid regions (dotted area, AI > 1), where ecosystems are unlikely to be water limited, the impact of $\Delta\theta_{crit}$ is likely to be negligible. **b**, The colours are mapped along the two axes representing the absolute changes in $\theta_{crit}$ (*y* axis) and relative changes in potential transpiration rate ($\Delta T_{pot}$, *x* axis), respectively. Each pixel is mapped continuously in its opacity, from transparency (0% change) to full intensity (99% of all observations), while the colours change continuously from sand (red) to clay (yellow) based on the soil-texture-specific relationship between $\Delta T_{pot}$ and $\Delta\theta_{crit}$ (the colours stem from the 12 different soil textural classes; compare also the different slopes, for example, clay versus sand, to Fig. 1c). Warm colours (red–orange–yellow) indicate an increase (+$\Delta\theta_{crit}$) and cold colours (blue–green) indicate a decrease (−$\Delta\theta_{crit}$) in critical soil moisture thresholds.

Note that our analysis was based on state variables (transpiration as a function of soil moisture and VPD) and did not consider the temporal scale—that is, how quickly the soil dries and how frequently ecosystems are water or energy limited. The temporal dynamics of soil moisture are important and influenced by soil properties in several ways. For example, soil properties regulate how water infiltrates, how soils are drained and how much water is available to plants. Addressing the temporal dynamics of soil drying would include an analysis of how plant hydraulics, leaf area and root depth change and potentially adapt to the local climate and soils on a seasonal time scale[37].

## Ecosystem water limitation under climate change

The dependence of critical soil water thresholds on soil hydraulic properties also indicates that climate change impacts on ecosystem water limitation will be modulated by soil texture. The majority of climate projections suggest a widespread increase in VPD[8,9,38], which will result in an increase in potential transpiration rate (+$\Delta T_{pot}$)[39]. Therefore, an increase in VPD is expected to cause an increase in $\theta_{crit}$ (+$\Delta\theta_{crit}$), as supply-limited flux conditions are reached at higher soil moisture (Extended Data Fig. 2a). Under these conditions, ecosystems will become water-limited earlier (+$\Delta\theta_{crit}$), that is, flux downregulation at a higher soil moisture, during seasonal soil drying, with potential negative effects on vegetation, such as reduced GPP. Contrastingly, the onset of water limitation at higher soil moisture suggests an earlier downregulation of plant water use (Extended Data Fig. 2b)—a mechanism that saves water and potentially delays the risks of severe water stress and drought mortality (Extended Data Fig. 2c). The slope of the relationship between $\theta_{crit}$ and $T_{pot}$ is soil-texture-dependent, with $\theta_{crit}$ ($T_{pot}$) being steeper in fine- than coarse-textured soils (Fig. 4b), which is explained by the hydraulic conductivity curves of the respective soils. It follows that critical soil water thresholds in fine-textured soils are comparatively more sensitive to VPD than in coarse soils. The soil texture modulation of $\Delta\theta_{crit}$ may thus have manifold implications for plant functioning and ecosystem fluxes. In the following, we investigate how soil texture would globally mediate the effect of $\Delta T_{pot}$ on the onset of ecosystem water limitation, that is, $\theta_{crit}$.

Globally, the average $\Delta\theta_{crit}$ is predicted to change by +0.004 from current (2005–2014) to future (2060–2069) climate (Shared Socio-economic Pathways (SSP) 2-4.5 scenario). This average change, and even the entire range (−0.003 to +0.012, equal to less than 10% relative change), are remarkably small compared to the $\Delta T_{pot}$ (from −19% to +65%) and the absolute values of $\theta_{crit}$ (from less than 0.1 to about 0.3). The small $\Delta\theta_{crit}$ means that the onset of ecosystem water limitation is only marginally sensitive to changes in VPD. This is explained by the steepness of the soil hydraulic conductivity curves, which causes a large drop in conductivity for a small change in soil moisture.

The regions with the largest +$\Delta\theta_{crit}$ were found in fine-textured soils with ecosystems responding most sensitively and promptly to future evaporative demands. In other words, ecosystems in fine-textured soils are expected to show the most pronounced effects of earlier stomatal downregulation. In relatively humid regions undergoing only periodic water limitation, this may result in GPP loss or may offset the positive effect of rising temperatures on vegetation growth in temperature-limited ecosystems[40,41] (Extended Data Fig. 3). In drier climates, this may instead help save water for periods of severe drought stress, although an acceleration of the water cycle (that is, an acceleration of seasonal soil drying) can be expected in many regions due to widespread increases in VPD and $T_{pot}$, respectively (Extended Data Fig. 4b).

Contrastingly, a small +$\Delta\theta_{crit}$ despite a substantial increase in VPD can be found in coarse-textured soils (red regions). In these regions, the effect of earlier stomatal closure is strongly diminished. Therefore, ecosystems in coarse-textured soils (highlighted by red rectangles) subject to regular water limitation may experience a high risk of exacerbated water stress and drought mortality (Fig. 4 and Extended Data Fig. 3). Notably, this includes some regions that are expected already or in the future of being vulnerable to drought, such as the American Southwest and southern Amazonia[42–44].

This simplified analysis of climate change impacts on terrestrial ecosystems only considers the effect of VPD on critical soil water thresholds. In fact, climate change impacts on vegetation and ecosystems depend on many more factors, such as changes in other components of the hydrological cycle[38,45], plant species composition, hydraulic

diversity and ecosystem resilience to drought[46–48], future temperature[49] and carbon dioxide ($CO_2$)[45]. Nonetheless, our analysis shows the effects of soil texture on ecosystem water limitation and should be considered so as to better understand the impacts of climate change on terrestrial ecosystems. When included in the comprehensive modelling of global land–atmosphere dynamics, our results may help to improve the management of drought risk under future climate scenarios.

## The central role of soil texture

Global ecosystem-scale observations, coupled with principles of water flow in soils and plants, show how soil and plant hydraulic conductivities determine the transition from energy to water limitation in terrestrial ecosystems. The dependence of critical soil water potential on soil texture underlines the role of soil hydraulics on ecosystem water limitation globally. The implications are manifold. First, the relative sensitivity of ecosystems to VPD and soil moisture depends on soil texture. Consequently, the predicted changes in $\theta_{crit}$ for future climates (that is, changes in potential transpiration rates via changes in VPD) are soil-texture-dependent. Given the limited adaptability of plants to critical soil water thresholds, a widespread increase in VPD may thus exacerbate water stress in many ecosystems more than previously assumed. Second, the extent to which plants can shape ecosystem water limitation by adjusting their hydraulic traits (for example, $K_{plant}$ and $\psi_{x*}$) depends on the soil texture and is particularly limited in sandy soils.

Plant adjustments that alter the soil hydraulic properties adjacent to the roots (the rhizosphere) are effective in weakening a drop in hydraulic conductivity (particularly relevant in soils with steep hydraulic conductivity curves, such as sand—note the sensitivity to $L_{root}$ in Fig. 2c). Plants have developed several strategies to enhance their ability to acquire water from the soil, such as the growth of root hairs[50], the exudation of polymers, such as mucilage[51], the symbiosis with mycorrhiza[52,53], and soil water-sensing strategies, such as hydropatterning[54]. Root and rhizosphere plasticity have also been reported in the context of variations in soil properties. Plants grow more and thicker roots in sandy soils[55], and have denser and longer root hairs in these soils[56,57]. Therefore, we expected that our model predictions, which assumed identical plant traits across soils, would have overestimated $\theta_{crit}$ in coarse-textured soils and underestimated it in fine-textured soils. Because this was not the case—in fact, it was rather the opposite—it suggests the occurrence of additional limitations in coarse-textured soils, such as loss of root-to-soil contact[58].

Over time, plants and microorganisms change the soil structure, with effects on soil water dynamics. Therefore, our simplified analysis, based on texture and sand content, should be further developed to include soil structure, and its dynamics and feedback with the vegetation and soil biota. In conclusion, we argue that accurate measurements and representations of soil and rhizosphere hydraulic properties are essential for predicting site-specific critical soil water thresholds. In particular, the heterogeneity of soil hydraulic properties due to the dynamic interactions between texture and soil formation processes should be addressed in a new quantitative framework that includes soil structural properties[59,60]. Rather than only using pedotransfer functions, which introduce additional prediction uncertainty, we advocate measuring soil hydraulic properties locally, especially the unsaturated soil hydraulic conductivity.

We have demonstrated the global importance of soil hydraulic properties in shaping ecosystem water limitation under current and future evaporative demands, and have provided new insights into ecosystem drought responses. Soil hydraulic properties are likely to influence many aspects of current and future terrestrial ecosystem functioning, such as drought-induced vegetation mortality[61], ecosystem resilience to drought[46–48] and climate extremes through land–atmosphere feedbacks[62,63]. Therefore, predictions of the terrestrial water cycle, the land carbon sink and ecosystem sensitivity to VPD versus soil moisture should include more detailed information on soil hydraulic properties. Furthermore, the global relevance of soils will increase in the future because ecosystems are expected to shift widely from energy to water limitation[34]. Overall, we recommend taking a deeper look at the hidden half of terrestrial ecosystems, given its large influence on ecosystem water limitation globally. A better understanding and parameterization of the mechanisms affecting ecosystem water limitation may ultimately help to safeguard vital ecosystems that are vulnerable to drought, not least under future climate conditions.

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

**Publisher's note** Springer *Nature* remains neutral with regard to jurisdictional claims in published maps and institutional affiliations.

## Methods

### Linking stomatal regulation and soil–plant hydraulics

As the soil dries, its water potential, $\psi$ (which is the negative work to extract a unit of volume of water), and its hydraulic conductivity, $k$, decrease by several orders of magnitude. Therefore, there is a critical soil water threshold (moisture or potential) at which the soil can no longer supply water to the roots at the rate required to sustain the transpiration demand[29,64]. The relationship between water supply and demand provides the mechanistic link between soil–plant hydraulics and stomatal regulation in water and carbon exchange between vegetation and atmosphere[20,25]. The nonlinear decrease in soil–plant hydraulic conductance with soil drying or increasing VPD has the key role in this framework. Note that different definitions and meanings of hydraulic 'conductance' and hydraulic 'conductivity' may create confusion. Here we refer to hydraulic conductance, $K$, as the ratio between the water flow, such as $T$ (millimetres per day), and the water potential difference, $\Delta\psi$ (megapascal), yielding millimetres per day per megapascal, while we refer to soil hydraulic conductivity, $k$ (metres per second), to the ratio of the water flux density, $q$ (metres per second), to the gradient in soil water potential along the flow path, $d\psi/dx$ (metres per metre). It is believed that stomata downregulate transpiration and photosynthesis at the point where transpiration comes at a disproportional 'cost' to water transport[20,25,30,35]. Despite the popularity and mechanistic strength of the framework, and models derived from this, it remains unclear which is the limiting hydraulic element of the soil–plant continuum. There is also discrepancy in the rules applied to derive stomatal regulation from the disproportionate cost of water transport (for example, centred on stomatal optimality, hydraulic conductance or physiological mechanisms). Here we used the supply–demand framework, implemented according to ref. 25.

### General model description

Critical soil water thresholds were calculated based on the hydraulic framework formulated in ref. 25. The premise is that stomata downregulate transpiration when the relationship between transpiration and leaf water potential becomes nonlinear (that is, loss in hydraulic conductance). The onset of nonlinearity is the uppermost limit of transpiration that can be supplied by soil–plant water flow before stomatal closure restricts the transpiration rate and photosynthesis. Hence, $\theta_{crit}$ and $\psi_{crit}$ are defined as the minimum soil moisture and water potential at which the soil–plant hydraulic system can supply water at the rate of the potential transpiration ($T_{pot}$). In other words, $\theta_{crit}$ and $\psi_{crit}$ are defined where $T_{pot}$ (here, 4 mm d$^{-1}$) is at the edge of the linear zone of the $T(\psi_{soil}, \psi_{leaf})$ surface (green zone in Extended Data Fig. 1a,b) intersecting the stress onset limit (SOL)[25].

The surface $T(\psi_{soil}, \psi_{leaf})$ is the physical space of plant water use, and the SOL delineates between the linear and nonlinear zones (yellow in Extended Data Fig. 1a,b), thereby defining the onset of water limitation[25]. In wet soils, the surface is planar for a large range of transpiration rates (green zone). As the soil dries, the surface bends (brown zone) as the relationship between transpiration rate and leaf water potential for a given soil water potential becomes nonlinear (dotted black lines in Extended Data Fig. 1a). This nonlinearity corresponds to a substantial decline in the hydraulic conductance of the soil–plant continuum. The transition from the linear to the nonlinear zone—the SOL—is defined as the point where $dT/d\psi_{leaf}$ reaches 80% of its maximum (for each $\psi_{soil}$). This indicates a substantial loss of hydraulic conductance in the soil–plant continuum and is hypothesized to trigger stomatal closure[25]. The SOL presumes a plant physiological optimization to minimize the trade-offs between gas-exchange benefits and hydraulic losses. The SOL is the uppermost limit of transpiration. It sets the maximum transpiration and corresponding stomatal conductance that plants could sustain under given soil water and VPD conditions. Obviously, stomatal conductance can be lower than this value, for

instance, limited by light and elevated $CO_2$. In other words, our model does not aim to reproduce stomatal closure driven by factors other than hydraulic limitation, which are crucial in predicting stomatal functioning below the SOL.

The model assumes that stomata progressively close when the hydraulic supply does not match the water demand—a process that has a time scale of minutes to hours. We used this model to predict the onset of water limitation during soil drying, a process that is comparatively slower and has a time scale of weeks. During this time, the plant hydraulics may change (particularly for grasses and annual crops). A key assumption of our analysis was that the relevant hydraulic variables changed proportionally, that is, $T_{pot}$, $K_{plant}$ and $L_{root}$ were assumed to change proportionally.

### Modelling steps and parameter estimation

To test our hypothesis, that critical soil moisture thresholds show soil texture specificity on an ecosystem scale, we simulated soil water thresholds for each soil textural class (12 US Department of Agriculture (USDA) classes) and compared them to both flux-tower and sap flow-derived observations. We simulated $\theta_{crit}$ and $\psi_{crit}$ by solely varying the soil hydraulic properties (Supplementary Table 2), while keeping plant and climate parameters constant (Extended Data Table 1). We justified the constant set of plant and atmospheric model parameters using the insignificant relationships of differences between observed and simulated $\theta_{crit}$ to site-specific latent heat fluxes (that is, to the absolute evapotranspiration rates determined by the climate of each site, $T_{pot}$) across climates and biomes (Supplementary Figs. 5 and 6). The few required model parameters were set to average values from the literature, except for the effective $L_{root}$. The potential transpiration per land surface, $T_{pot}$, was set to 4 mm d$^{-1}$ (during daytime)[65]. The maximum plant hydraulic conductance was set as $K_{plant\text{-}max} = T_{pot}/-\psi_{leaf\text{-}max}$ (mm d$^{-1}$ MPa$^{-1}$), which gave a leaf water potential of −1 MPa ($\psi_{leaf\text{-}max}$) when the soil was wet ($\psi_{soil} \approx 0$ MPa) and transpiration was at a maximum, that is, $T_{pot}$. The plant water potential threshold ($\psi_{x^*}$) at which the stem hydraulic conductance is reduced to 50% (approximately $\psi_{x50}$) was set to −2.8 MPa, based on reported values of xylem embolism in woody species[66], while the effective root and rhizosphere radii, and the slope of the decrease in $K_{plant}$ with increasing water tension, were set to default values[25]. The effective $L_{root}$ (m m$^{-2}$), defined per land surface area, was the only fitting parameter, and was inversely estimated by fitting $\theta_{crit}$ over all soil textural classes. In other words, the difference between observed ($\theta_{obs}$) and simulated ($\theta_{sim}$) critical soil moisture was minimized over both datasets across all soils (least absolute deviations) by varying only $L_{root}$ (where $n = 149$ is the number of $\theta_{crit}$ observations).

$$L_{root} := \min \sum_{i=1}^{n} |\theta_{i,\mathrm{obs}} - \theta_{i,\mathrm{sim}}|$$

### Main data collection

To test our hypotheses, we compared our model simulations to both eddy covariance and sap flow data across climates and biomes. To estimate critical soil moisture thresholds ($\theta_{crit}$) from the eddy covariance data, we acquired daily data comprising the soil volumetric water content and the latent and sensible heat fluxes from the eddy covariance sites provided by Integrated Carbon Observation System (ICOS) (https://www.icos-cp.eu/), AmeriFlux (https://ameriflux.lbl.gov/) and FLUXNET (https://fluxnet.org/), all of which have undergone standardized quality control and gap filling[67] (we used both measured, quality flag = 0, and good-quality gap-filled, quality flag = 1, data). Only sites for which in situ estimates of soil texture were available were selected ($n = 44$). These sites either reported the soil textural class or provided the fractions of sand, silt and clay from which we could classify the soil texture based on the USDA soil texture classification system[68]. Given the high correlation of critical soil moisture thresholds in the surface

soil layer with the $\theta_{crit}$ observed in deeper layers[36], only the surface layer soil moisture was considered in our analysis. To estimate critical soil moisture thresholds ($\theta_{crit}$) using sap flow data, we acquired the sapwood-area-based sap flux density ($cm^3 \, cm^{-2}_{Asw} \, h^{-1}$) and soil volumetric water content time series from SAPFLUXNET[69] (https://sapfluxnet.creaf.cat/) using the provided sapfluxnetr package v.0.1.4 (ref. 70). Only sites that reported local estimates of $\theta$ (shallow soil layer), soil texture (USDA classification, and/or sand, silt and clay fractions) and soil depth along the sap flux density values were kept for further analysis.

## Main data analysis

**Eddy covariance data.** The evaporative fraction was calculated as the ratio of the latent heat flux to the sum of the latent and sensible heat fluxes for each day[71,72]. $\theta_{crit}$ was determined from the relationship of the evaporative fraction to $\theta$ by applying a regression between evaporative fraction and $\theta$ using a linear-plus-plateau model (lin_plateau.R function from ref. 73). The good correlation between the onset of a decline in evaporative fraction and a decline in gross primary production[4] justified the approach to interpret these data as a decline in transpiration, although bare evaporation can substantially contribute to the evaporative fraction. $\theta_{crit}$ is defined as the soil moisture at the breakpoint between the linear increase phase and the plateau of the model. As in ref. 36, $\theta_{crit}$ was estimated from periods of soil moisture dry-downs during the respective summer seasons (that is, June–July–August for the Northern Hemisphere and December–January–February for the southern hemisphere). Soil moisture dry-downs were considered to be periods in which the soil moisture decreased consecutively for at least 10 days after a rain event[33,74]. We could determine $\theta_{crit}$ for 36 out of the 44 sites (see 'Main data collection') using the dry-down definition from ref. 36. For five sites, we determined $\theta_{crit}$ for periods beyond the summer season, but still applying the 10-day drying criterion. For two sites, this dry-down criterion also did not result in a $\theta_{crit}$ estimate, and thus we neglected the 10-day dry-down criterion, rather applying the summer criterion. Finally, we discarded one site for which no $\theta_{crit}$ could be determined at all. The remaining 43 sites formed the basis for all further analysis. We justified the different dry-down criteria by the high coefficients of determination ('Summer' versus 'Full-criterion': $R^2_{adj} = 0.97$, $n = 11$; '10 days' versus 'Full-criterion': $R^2_{adj} = 0.98$, $n = 10$). The eddy covariance sites span around the globe, encompassing all continents (excluding Antarctica) (Supplementary Fig. 7).

**Sap flow data.** Similarly to ref. 75, (sub)hourly sap flux density time series were aggregated to daylight averages (06:00 to 20:00) using daylight_metrics from sapfluxnetr[70]. As for the eddy covariance data, we estimated dry-down periods as periods where the daily (24 h) averages of $\theta$ decreased for at least ten consecutive days (site level). After intersecting summer and dry-down periods, $\theta_{crit}$ was determined as for the eddy covariance data, but on a tree level (multiple trees in the site sharing the same environmental data) using the linear-plus-plateau model (now part of the soiltestcorr package v.2.2.0 (ref. 76)), given that a positive linear slope was determined and that the breakpoint determination met the standard significance criterion ($P < 0.05$). Finally, 14 sites (Supplementary Fig. 8) and 106 trees (multiple tree individuals per site, either from the same or a different tree species) resulted in 106 sap flow-derived estimates of $\theta_{crit}$, spanning six soil textural classes (Supplementary Table 1).

## Soil hydraulic properties

The parameters of the soil water characteristics as a function of soil textural classes were taken from ref. 77, which reported the mean and standard deviation. Because the saturated conductivity data in ref. 77 were from another data source and without information on its variability, we took the values from ref. 78 that provided a recent global data collection.

## The relative importance of soil and plant hydraulics

Analysing the relative importance of soil and plant hydraulics is key to identifying the dominant controls on ecosystem water limitation. We approached this in two ways: (1) by comparing the simulated soil and plant hydraulic conductance as a function of soil texture (Fig. 1b); and (2) by comparing the differences in $\theta_{crit}$ variability between observations and simulations (Fig. 2c).

Simulating the physical space and transpiration downregulation (SOL) in each soil textural class allowed us to disentangle, by means of the soil–plant hydraulic model, whether the soil or plant hydraulics would have, in relative terms, a stronger impact on soil water thresholds. We calculated the soil and plant hydraulic conductance as $K_{soil} = T/(\psi_{soil} - \psi_{soil\text{-}root})$ and $K_{plant} = T/(\psi_{soil\text{-}root} - \psi_{leaf})$, respectively, where $\psi_{soil\text{-}root}$ and $\psi_{leaf}$ are water potentials at the soil–root interface and in the leaves, respectively. To analyse the $\theta_{crit}$ variability, we quantified the variation in $\theta_{crit}$ in response to variations in soil hydraulic properties for each soil textural class. First, we determined, for each soil textural class, the minimum and maximum values of a hydraulic property by subtracting or adding the standard deviation from the mean value (the geometric mean, in the case of saturated hydraulic conductivity and the shape parameters of the soil water characteristics curve). Next, we defined the hydraulic properties of the 'coarse end' of a soil textural class by combining the minimum air entry value, maximum slope parameter of soil water characteristics curve and maximum hydraulic conductivity for each soil textural class. For the 'fine end' of a soil textural class, the maximum air entry value, minimum slope parameter and minimum soil hydraulic conductivity values were chosen. Note that $\tau$, which is the slope of $k_{soil}$ over $\psi_{soil}$ when both are expressed in logarithmic scale, is positively correlated with $K_{sat}$ and inversely correlated with the air entry value, $h_b$ (the correlations between $\log(h_b)$ and $\tau$ and $\log(K_{sat})$ and $\tau$ in the data presented in ref. 77 are 0.78 and 0.88, respectively). To test the effects of variable plant traits ($L_{root}$ density and plant vulnerability) and atmospheric conditions (increasing VPD) on soil water thresholds, we modelled $\theta_{crit}$ and $\psi_{crit}$ by varying $L_{root}$ from 1/30 (minimum) to 30 (maximum) times the reference, and $\psi_{x*}$ from −1.5 (minimum) to −5 MPa (maximum). From these results, we calculated the span of $\theta_{crit}$ ($S_{\theta crit}$, unitless) for each soil textural class as $S_{\theta crit} = \max(\theta_{crit}) - \min(\theta_{crit})$, which allowed us to compare the effects of varying soil and plant properties to the variance of the observations—that is, the span of the FN and SFN observations per soil textural class. Varying atmospheric conditions were simulated using future projections of potential transpiration rates. Details of the future climate modelling are described in the section 'Global map'.

## The relative importance of VPD and soil moisture in ecosystem water limitation

To evaluate the relative importance of VPD and soil moisture in ecosystem water limitation, we compared the soil-specific simulations with observed ecosystem fluxes for two contrasting soil textures (median of five eddy covariance sites for clay and sand, respectively). For these simulations, we assumed that $T_{pot} = 4$ mm d$^{-1}$, corresponding to VPD = 1.5 kPa (Fig. 3a–d). Our model predicted that, at VPD = 1.5 kPa, plants could transpire at full stomatal opening ($g_{cmax}$) as long as the soil moisture was higher than, or equal to, the simulated $\theta_{crit}$ (that is, moving horizontally in Fig. 3). Rising VPD (that is, moving vertically in Fig. 3) triggered stomatal closure at a critical VPD, which was set by the stress onset limit (yellow line in Extended Data Fig. 1). The critical VPD declined with decreasing soil water content, but it remained relatively constant for $\theta > \theta_{crit}$ (particularly in sandy soils; in Fig. 3, this critical VPD is approximately 2 kPa). This critical VPD in wet soil depends on plant hydraulics and $T_{pot}$. More precisely, critical VPD depends on the difference between $T_{pot}/K_{plant}$ and the critical leaf water potential where $K_{plant}$ declines (psi_star, Supplementary Information). For instance, transpiration would become water-limited at a low VPD (and high soil

moisture) when the plant hydraulic conductance was too low to sustain high transpiration fluxes. In this case, the system becomes water limited even in wet soils due to plant limitations, the key driver being the rising VPD.

For the observed evaporative fractions in clay and sand, we used (half-)hourly fluxes from these eddy covariance towers and filtered them as follows, referring to previous studies[1,2], to remove unmeaningful data for our analysis: only positive latent and sensible heat fluxes; only during daytime; without negative soil moisture and VPD values; sufficient incoming radiation and VPD to drive substantial transpiration (photosynthetic photon flux density of more than 500 $\mu$mol m$^{-2}$ s$^{-1}$, VPD of more than 0.5 kPa); sufficient wind speed (more than 1 m s$^{-1}$) to foster vegetation–atmosphere coupling; and without 'cold' days limiting plant metabolism (where the median daily temperature was less than 15 °C). Because we aimed to analyse as much of the VPD–soil moisture space as possible, we used the maximum available time resolution (half-hourly to hourly) and all levels of gap-filled data. Binning and visualization of the evaporative fraction along VPD and soil moisture (thereby removing soil moisture values below and above the residual and saturated water content of each soil texture, respectively, as well as cutting off the extreme VPD conditions (VPD of more than 5 kPa) occurring in one sand site) revealed soil-specific responses to the two environmental drivers. Simulations of transpiration rate with soil drying, based on the SOL in each of the two soil textures, were anchored to the VPD axis by the assumption that the maximum transpiration rate being sustained by the underlying soil–plant hydraulic constraints corresponded to the experimentally observed maximum evaporative demand (we took the 99th percentile of the median VPD distribution of the five sites) in wet soil ($\theta > \theta_{crit}$) for each textural class. Inset plots displaying the median relative sensitivity of evaporative fraction to VPD and $\theta$ confirmed the stronger relative contribution of soil moisture limitation in coarse-textured soils.

## Global map

To test the effects of variable atmospheric conditions and evaluate the impacts of future climate on $\theta_{crit}$, we simulated $\theta_{crit}$ in all soil textural classes under changing potential transpiration demands. For each soil textural class, we obtained a numerical function of $T(\theta)$, as in Supplementary Fig. 9 (note that the slightly different model parameters enabled projections of future transpiration rate up to the maximum increase in $T_{pot}$, that is, +65%). An analytical sigmoidal function was fitted to $T(\theta)$ to calculate $\theta_{crit}$ and $\Delta\theta_{crit}$ for the expected changes in potential transpiration rate under future climate (2060–2069) (Fig. 4 and Extended Data Figs. 3 and 4). To estimate the potential transpiration rate, we chose a simple approach based on air temperature and relative humidity, as described by Ivanov's formula[79,80]. Temperature and relative humidity data for the years 2005–2014 (current climate) and 2060–2069 (future climate) were downloaded from World Climate Research Programme Coupled Model Intercomparison Project 6 (SSP2-4.5 scenario) using the EC-Earth3 model with a spatial resolution of 0.7° (refs. 81,82). Additionally, current and future precipitation data were acquired to classify world regions based on the current and future AI. The AI was calculated on an annual basis as AI = precipitation/$T_{pot}$. Next, the global map of $\Delta\theta_{crit}$ resulting from changing climate ($T_{pot}$) was calculated as follows: in a first step, global maps of sand and clay content from SoilGrids[83] were used to determine a global map of soil textural classes (Extended Data Fig. 4a). For each pixel, the change in potential transpiration rate (Extended Data Fig. 4c) and the corresponding change in critical water content were then computed and visualized (Fig. 4 and Extended Data Figs. 3 and 4).

## Statistical information

Functional relationships between key variables were underpinned by linear regression analyses using 'lm()' from the stats package

(v.4.3.2) of R statistical software v.4.3.2 (ref. 84). The linear regression prerequisites were verified using 'check_model()' from the performance package (v.0.11.0) (ref. 85) (see 'Code availability' for details). As a goodness of linear regression fit, adjusted $R^2_{adj}$ values and two-sided $P$ values of the linear regression slopes were calculated and are indicated in the figures, where meaningful (significant $P$ values are indicated in the plot area using standard significance levels of *$P < 0.05$, **$P < 0.01$, ***$P < 0.001$, not significant (NS) $P > 0.05$). The main linear regressions, indicated by $R^2_{adj}$ and significance levels in the main text figures, are detailed as follows: Fig. 2a, solid black regression line, $R^2_{adj} = 0.34$, $P < 0.001$, $y = -0.0017x + 0.23$; Fig. 2b, solid black regression lines encompassing all textures, $R^2_{adj} = 0.35$, $P < 0.001$, $y = 0.61x + 0.05$, and excluding clay soils, $R^2_{adj} = 0.48$, $P < 0.001$, $y = 0.98x + 0.01$; Fig. 2c, solid black regression line, $R^2_{adj} = 0.4$, $P < 0.01$, $y = -0.0019x + 0.23$, dotted brown regression line, $R^2_{adj} = 0.87$, $P < 0.001$, $y = -0.0018x + 0.24$, dotted dark green regression line, $R^2_{adj} = 0.87$, $P < 0.001$, $y = -0.0013x + 0.15$; Fig. 2d, solid black regression line, $R^2_{adj} = 0.37$, $P < 0.001$, $\log_{10}(y) = -0.048x + 2.6$ (see 'Code availability' for further details). Note that, in Fig. 2b, we tested whether the 95th intervals of the estimated slopes and intercepts of the linear regressions overlapped with the 1:1 line (see 'Code availability' for details). The error bars for the sand fraction (in Fig. 2a) show the entire range of sand percentages within each soil textural class[68]. The data distributions are displayed using grouped box plots (the thick solid horizontal line represents the median, the lower and upper hinges correspond to the first and third quartiles, the whiskers extend to the highest or lowest value, respectively, but no further than 1.5 times the interquartile range, and the widths of the boxes scale with the square root of the number of observations in each soil textural class) and individual observations in the groups are displayed as points along the boxes. Note that the grouped (FN and SFN) boxes and points (Fig. 2a,b) are slightly shifted around the true $x$ coordinate (the same for both) for readability. The number of sites in each soil textural class is given in Supplementary Table 2, while the number of $\theta_{crit}$ observations per soil textural class are displayed in Fig. 2a,b.

## Reporting summary

Further information on research design is available in the Nature Portfolio Reporting Summary linked to this article.

## Data availability

All ecosystem flux data are publicly available from FLUXNET (https://fluxnet.org/), AmeriFlux (https://ameriflux.lbl.gov/) and the ICOS (https://meta.icos-cp.eu/collections/ueb_7FcyEcbG6y9-UGo5HUqV). Additional soil texture information, where missing, was kindly provided from scientists responsible for the respective eddy covariance site. All sap flow data are publicly available from SAPFLUXNET (https://sapfluxnet.creaf.cat/). An overview of the analysed sites, including data references, is provided in Supplementary Table 2. Temperature, relative humidity and precipitation data for the years 2005–2014 (current climate)[81] and 2060–2069 (future climate)[86] are publicly available from the World Climate Research Programme Coupled Model Intercomparison Project 6 (https://aims2.llnl.gov/search/cmip6/) and were downloaded using the EC-Earth3 model (SSP2-4.5 scenario) with a spatial resolution of 0.7° (data deposited at Figshare (https://doi.org/10.6084/m9.figshare.24138300)[87]).

## Code availability

The codes to analyse and visualize the data were written in R v.4.3.2 (ref. 84), MATLAB R2023a (ref. 88) and Mathematica v.13.0 (ref. 89). All codes essential for this analysis are available at Figshare (https://doi.org/10.6084/m9.figshare.24138300)[87].

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

**Acknowledgements** We thank all those responsible for the eddy covariance and sap flow sites we analysed for their commitment to these sites generating high-quality data. Thanks also to the FLUXNET, AmeriFlux, ICOS and SAPFLUXNET communities for acquiring, harmonizing and sharing their data.

**Author contributions** F.J.P.W., A. Carminati and M.J. conceived the study. F.J.P.W. led the data analyses with contributions from L.D., A. Carminati, P.L., M.J.B. and A. Cecere. F.J.P.W. ran the model, initially developed by A. Carminati and M.J. All authors contributed to the interpretation of the analysis and its implications. F.J.P.W. and A. Carminati drafted the manuscript, with contributions from all authors.

**Funding** Open access funding provided by Swiss Federal Institute of Technology Zurich.

**Competing interests** The authors declare no competing interests.

**Additional information**
**Correspondence and requests for materials** should be addressed to Mathieu Javaux or Andrea Carminati.

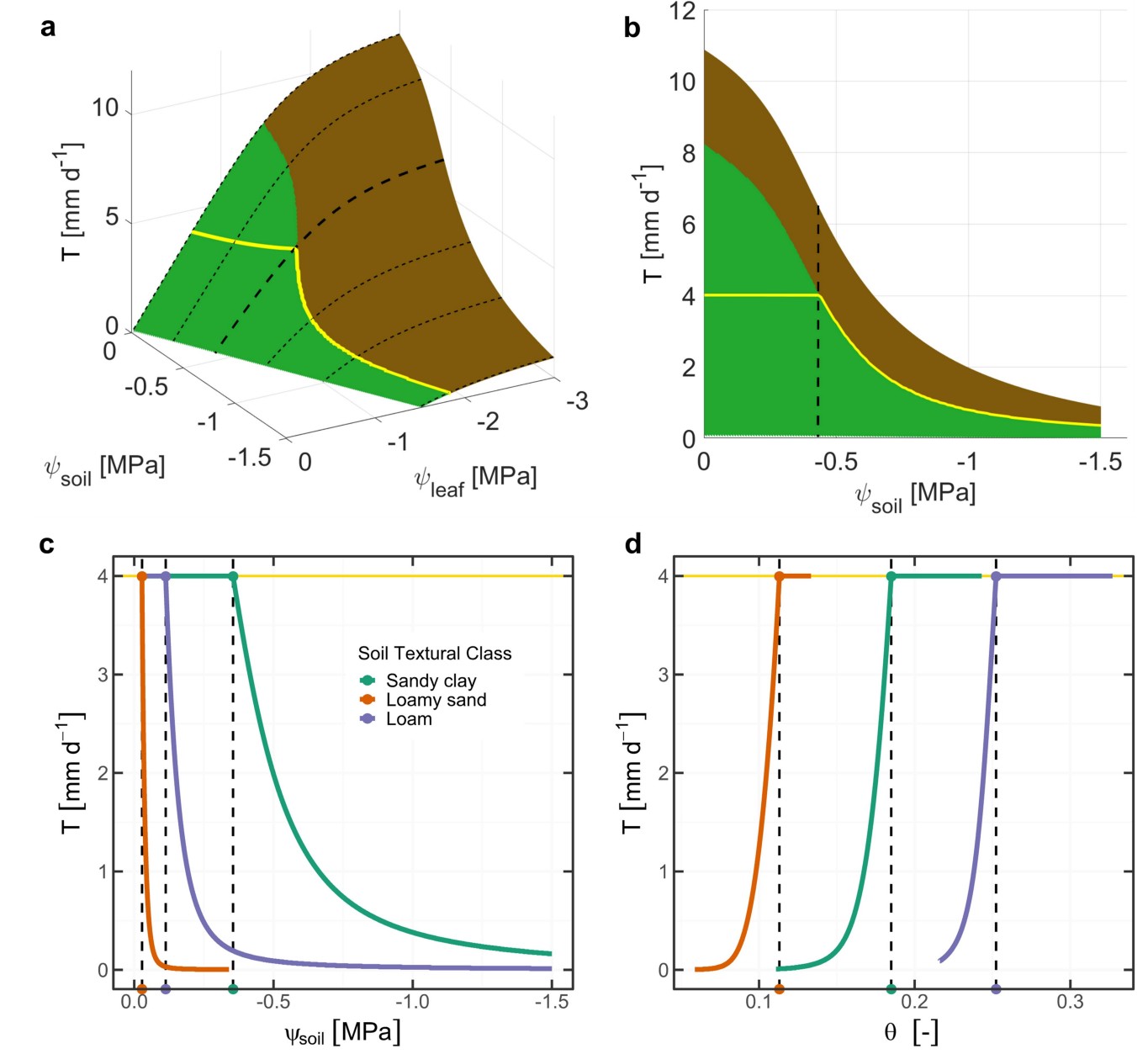

**Extended Data Fig. 1 | Coupling soil-plant hydraulics with stomatal regulation allows simulating soil-specific soil water thresholds. a**, Plant hydraulic surface and soil drying trajectory (thick yellow line) simulated[25] for exemplary hydraulic parameters and a potential transpiration rate of 4 mm/day. The relation is shown in **a** as 3-D hydraulic surface and in **b** from the soil point of view, distinguishing the linear (green) and nonlinear (brown) zones. **c,d** Exemplary land surface transpiration rate for three contrasting soil textural classes in relation to **c** bulk soil water potential and **d** volumetric soil moisture content. The three soils result in different $\theta_{crit}$ and $\psi_{crit}$ (dashed black lines) while the evaporative demand is the same (4 mm/day).

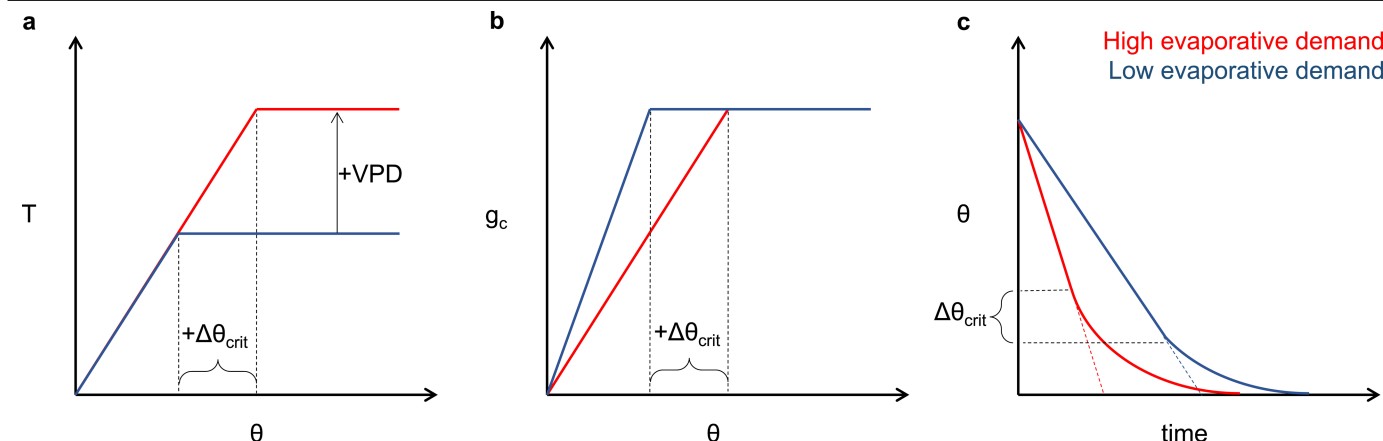

**Extended Data Fig. 2 | Expected changes of critical soil moisture thresholds in response to increasing VPD. a**, An increase in VPD (red) results in an increase in critical soil moisture threshold (+$\Delta\theta_{crit}$). **b**, An increase in VPD (red) causes an 'earlier' stomatal downregulation, i.e., at higher soil moisture. **c**, An increase in VPD (red) may dry the soil initially faster due to increase in transpiration rate in non-water-limited soil moisture (assuming no change in other components in the hydrological cycle), but the 'earlier' stomatal closure may prolong the time between onset of water limitation ($\theta_{crit}$) and severe water stress.

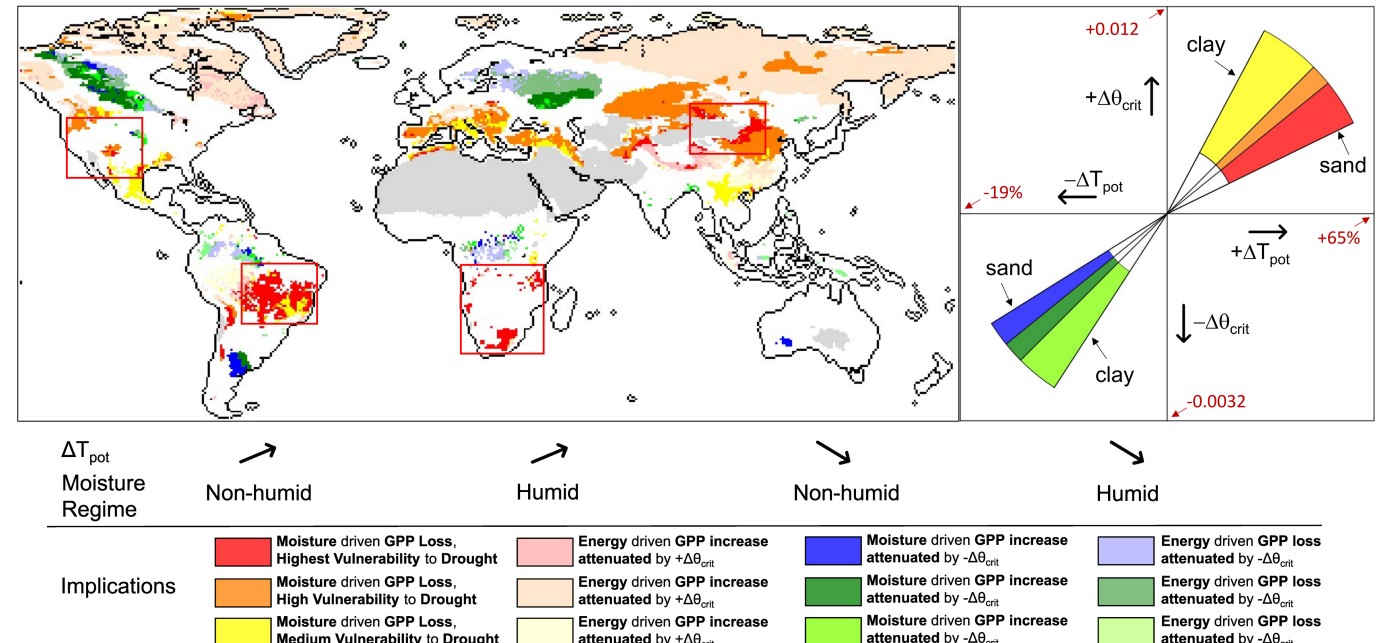

| ΔT_pot | | | | |
| Moisture Regime | Non-humid ↗ | Humid ↗ | Non-humid ↘ | Humid ↘ |
| Implications | Moisture driven **GPP Loss, Highest Vulnerability** to Drought (red) | Energy driven **GPP increase** **attenuated** by +Δθ_crit (pale red) | Moisture driven **GPP increase** **attenuated** by -Δθ_crit (blue) | Energy driven **GPP loss** **attenuated** by -Δθ_crit (pale blue) |
| | Moisture driven **GPP Loss, High Vulnerability** to Drought (orange) | Energy driven **GPP increase** **attenuated** by +Δθ_crit (pale orange) | Moisture driven **GPP increase** **attenuated** by -Δθ_crit (dark green) | Energy driven **GPP loss** **attenuated** by -Δθ_crit (green) |
| | Moisture driven **GPP Loss, Medium Vulnerability** to Drought (yellow) | Energy driven **GPP increase** **attenuated** by +Δθ_crit (pale yellow) | Moisture driven **GPP increase** **attenuated** by -Δθ_crit (light green) | Energy driven **GPP loss** **attenuated** by -Δθ_crit (pale green) |

**Extended Data Fig. 3 | The global sensitivity of critical soil moisture thresholds to climate change depends on soil texture – I.** Predicted changes of global critical soil moisture thresholds ($\Delta\theta_{crit}$) in response to changes in VPD from current (2005-2014) to future (2060-2069) climate (SSP2-4.5 scenario). The colors are mapped along two axes representing the absolute changes in $\theta_{crit}$ (y-axis) and relative changes in potential transpiration rate ($\Delta T_{pot}$, x-axis), respectively. Pixels are coloured across the globe according to their expected implications for gross primary productivity (GPP) and vulnerability to drought. Opaque and pale colours differentiate between non-humid (aridity index (AI) < 1) and humid (AI > 1) moisture regimes, respectively. Warm colours (red-orange-yellow) indicate an increase (+$\Delta\theta_{crit}$), and cold colours (blue-green) indicate a decrease (−$\Delta\theta_{crit}$) in critical soil moisture thresholds. The four rectangles highlight regions where we expect highest amplification of ecosystem vulnerability to drought given increasing VPD (cf. Fig. 4 in the Article). These regions will experience an increase in atmospheric drying but show limited buffer capacity (small $\Delta\theta_{crit}$) due to the coarseness of their soil texture. Hyperarid deserts (dark grey, aridity index (AI) ≤ 0.05) were excluded. In humid regions (pale colours) where ecosystems are unlikely to be water limited, the impact of $\Delta\theta_{crit}$ is likely to be negligible or rare.

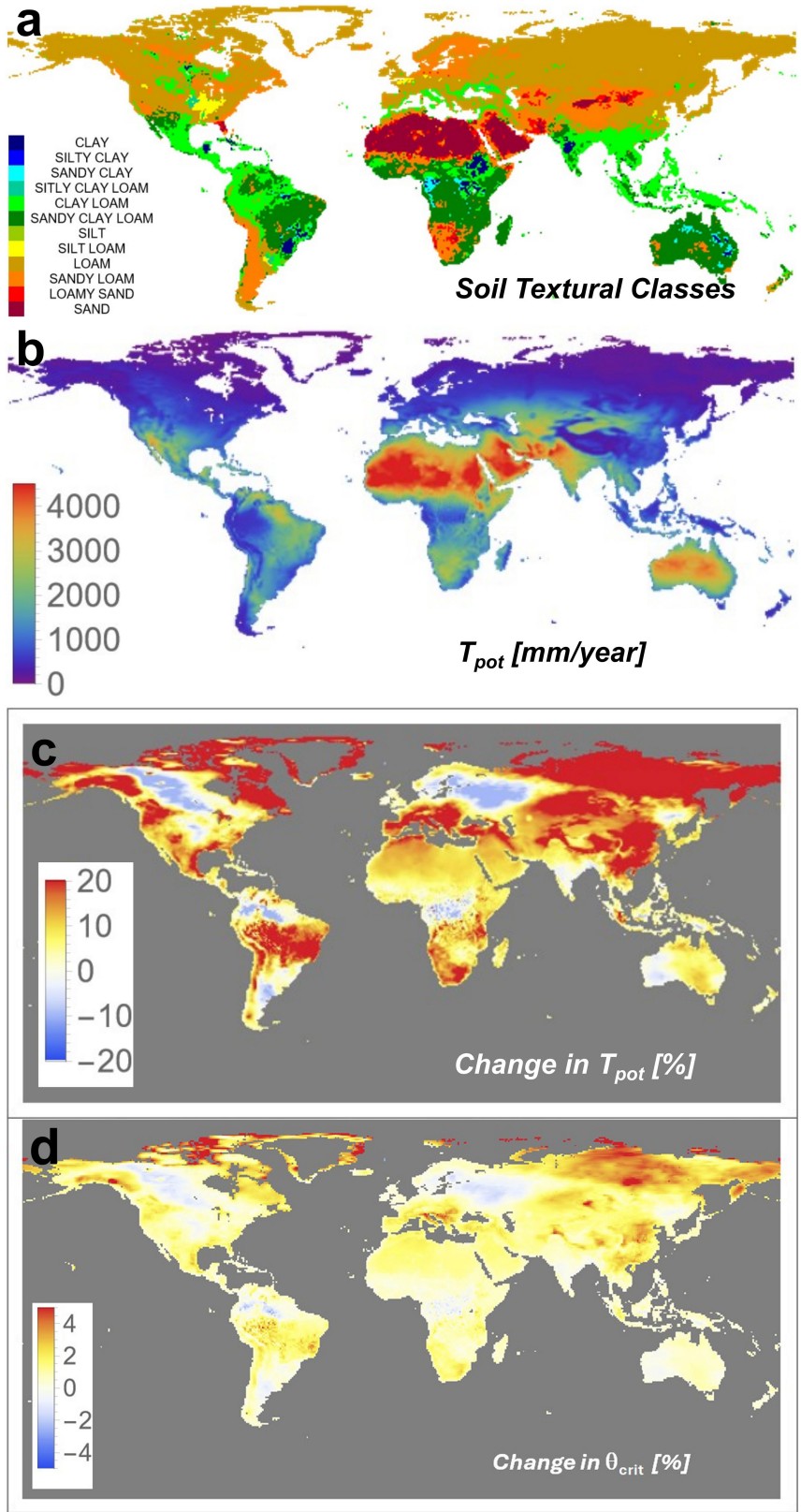

**Extended Data Fig. 4 | The global sensitivity of critical soil moisture thresholds to climate change depends on soil texture – II. a**, Global distribution of soil textural classes according to USDA classification determines how the projected changes in potential transpiration rate ($T_{pot}$) translate into changes in critical soil moisture threshold ($\Delta\theta_{crit}$) under future evaporative demand (2060-2069, cf. Fig. 4 + Extended Data Fig. 3). **b**, Current estimated annual $T_{pot}$ based on air temperature and relative humidity alone as described by Ivanov[79,80]. **c**, Estimated changes in $T_{pot}$ [%] under future evaporative demands driving the changes in $\theta_{crit}$ [%] in **d**. Note that here, in contrast to Fig. 4 + Extended Data Fig. 3, hyperarid and humid regions are not specifically marked, and the changes in $\theta_{crit}$ are relative, i.e., strongly depend on the soil textural class, which controls the absolute value of $\theta_{crit}$ [−].

## Extended Data Table 1 | Model Parameter Definitions and Derivations

| Parameter | Definition | Unit | (Default) Value | Reference/Derivation |
|---|---|---|---|---|
| $T_{pot}$ | Average Potential Transpiration per Land Surface Area | mm d$^{-1}$ | 4 | 65* |
| $\psi_{leaf-max}$ | Average Plant Water Potential at Full Transpiration ($T_{pot}$) | MPa | -1 | 88† |
| $R_{root}$ | Root Hydr. Resistance ($1/K_{plant-max}$) | MPa mm$^{-1}$ d | 0.25 | $-\psi_{leaf-max} / T_{pot}$ |
| $L_{root}$ | Effective Root Length | m m$^{-2}$ | 638 | Fitted to minimize the sum of absolute errors between observed and simulated $\theta_{crit}$ (see Methods) |
| $\psi_{x*}$ | Plant Water Potential Threshold ($\sim\psi_{x50}$) | MPa | -2.8 | 88‡ |
| $\tau_x$ | Plant Water Potential Exponent | | 5 | 25 |
| $r_0$ | Root Radius | cm | 0.05 | 25 |
| $r_2$ | Rhizosphere Radius | cm | 1 | 25 |

*$T_{pot}$ = 4 mm/d stems from the potential evaporation rate used in this study to derive physical constraints for soil evaporation globally.

†$\psi_{x-max}$ = −1 MPa represents approx. the median of reported leaf water potentials at full transpiration in wet soil contained in this database.

‡$\psi_{x*}$ = −2.8 MPa represents P50 given median values of the Weibull parameters (c and d, respectively) in this dataset.

# Reporting Summary

## Statistics

For all statistical analyses, confirm that the following items are present in the figure legend, table legend, main text, or Methods section.

| n/a | Confirmed | |
|---|---|---|
| ☐ | ☒ | The exact sample size ($n$) for each experimental group/condition, given as a discrete number and unit of measurement |
| ☒ | ☐ | A statement on whether measurements were taken from distinct samples or whether the same sample was measured repeatedly |
| ☐ | ☒ | The statistical test(s) used AND whether they are one- or two-sided<br>*Only common tests should be described solely by name; describe more complex techniques in the Methods section.* |
| ☒ | ☐ | A description of all covariates tested |
| ☐ | ☒ | A description of any assumptions or corrections, such as tests of normality and adjustment for multiple comparisons |
| ☐ | ☒ | A full description of the statistical parameters including central tendency (e.g. means) or other basic estimates (e.g. regression coefficient) AND variation (e.g. standard deviation) or associated estimates of uncertainty (e.g. confidence intervals) |
| ☐ | ☒ | For null hypothesis testing, the test statistic (e.g. $F$, $t$, $r$) with confidence intervals, effect sizes, degrees of freedom and $P$ value noted<br>*Give P values as exact values whenever suitable.* |
| ☒ | ☐ | For Bayesian analysis, information on the choice of priors and Markov chain Monte Carlo settings |
| ☒ | ☐ | For hierarchical and complex designs, identification of the appropriate level for tests and full reporting of outcomes |
| ☒ | ☐ | Estimates of effect sizes (e.g. Cohen's $d$, Pearson's $r$), indicating how they were calculated |

*Our web collection on statistics for biologists contains articles on many of the points above.*

## Software and code

Policy information about availability of computer code

| Data collection | N/A |
|---|---|
| Data analysis | The codes to analyze and visualize data were written in R (v4.3.2), Matlab (R2023a), and Mathematica (v13.0). All codes essential for this analysis are available at https://doi.org/10.6084/m9.figshare.24138300. R packages used, among others, include sapfluxnetr (v0.1.4), soiltestcorr (v2.2.0), stats (v4.3.2), and performance (v0.11.0). |

For manuscripts utilizing custom algorithms or software that are central to the research but not yet described in published literature, software must be made available to editors and reviewers. We strongly encourage code deposition in a community repository (e.g. GitHub). See the Nature Portfolio guidelines for submitting code & software for further information.

## Data

Policy information about availability of data

All manuscripts must include a data availability statement. This statement should provide the following information, where applicable:
- Accession codes, unique identifiers, or web links for publicly available datasets
- A description of any restrictions on data availability
- For clinical datasets or third party data, please ensure that the statement adheres to our policy

All ecosystem flux data are publicly available from FLUXNET (https://fluxnet.org/), AMERIFLUX (https://ameriflux.lbl.gov/) and ICOS (https://meta.icos-cp.eu/collections/ueb_7FcyEcbG6y9-UGo5HUqV). Additional soil texture information, where missing, was kindly provided from scientists responsible for the respective

## Research involving human participants, their data, or biological material

Policy information about studies with human participants or human data. See also policy information about sex, gender (identity/presentation), and sexual orientation and race, ethnicity and racism.

| | |
|---|---|
| Reporting on sex and gender | N/A |
| Reporting on race, ethnicity, or other socially relevant groupings | N/A |
| Population characteristics | N/A |
| Recruitment | N/A |
| Ethics oversight | N/A |

Note that full information on the approval of the study protocol must also be provided in the manuscript.

## Field-specific reporting

Please select the one below that is the best fit for your research. If you are not sure, read the appropriate sections before making your selection.

☐ Life sciences    ☐ Behavioural & social sciences    ☒ Ecological, evolutionary & environmental sciences

For a reference copy of the document with all sections, see nature.com/documents/nr-reporting-summary-flat.pdf

## Life sciences study design

All studies must disclose on these points even when the disclosure is negative.

| | |
|---|---|
| Sample size | N/A |
| Data exclusions | N/A |
| Replication | N/A |
| Randomization | N/A |
| Blinding | N/A |

## Behavioural & social sciences study design

All studies must disclose on these points even when the disclosure is negative.

| | |
|---|---|
| Study description | N/A |
| Research sample | N/A |
| Sampling strategy | N/A |
| Data collection | N/A |
| Timing | N/A |
| Data exclusions | N/A |
| Non-participation | N/A |
| Randomization | N/A |

# Ecological, evolutionary & environmental sciences study design

All studies must disclose on these points even when the disclosure is negative.

| | |
|---|---|
| Study description | Meta-analysis of published eddy-covariance and sapflux data collected together with local soil information (volumetric soil moisture, soil textural class/fractions of sand/silt/clay). Soil-plant hydraulic model used to test and explain global soil texture influence on observed critical soil moisture thresholds. |
| Research sample | All soil moisture as well as ecosystem and tree-level flux data were publicly available (see Manuscript). Additional soil texture information, where missing, was kindly provided from scientists responsible for the respective Eddy-Covariance site. |
| Sampling strategy | not applicable, no new data were recorded |
| Data collection | No new data were recorded, collected together. |
| Timing and spatial scale | No new data were recorded. Global scale |
| Data exclusions | All sites where soil texture was unambigously classified (locally) or fractions of sand/silt/clay were given were used. Further processing of this dataset (e.g. estimating critical soil water thresholds) is described in detail in the manuscript. |
| Reproducibility | no experiment done |
| Randomization | Sites were grouped according to their soil textural class. No randomization applicable. |
| Blinding | Sites were grouped according to their soil textural class. No randomization applicable. |

Did the study involve field work? ☐ Yes ☒ No

## Field work, collection and transport

| | |
|---|---|
| Field conditions | N/A |
| Location | N/A |
| Access & import/export | N/A |
| Disturbance | N/A |

# Reporting for specific materials, systems and methods

We require information from authors about some types of materials, experimental systems and methods used in many studies. Here, indicate whether each material, system or method listed is relevant to your study. If you are not sure if a list item applies to your research, read the appropriate section before selecting a response.

## Materials & experimental systems

| n/a | Involved in the study |
|---|---|
| ☒ | ☐ Antibodies |
| ☒ | ☐ Eukaryotic cell lines |
| ☒ | ☐ Palaeontology and archaeology |
| ☒ | ☐ Animals and other organisms |
| ☒ | ☐ Clinical data |
| ☒ | ☐ Dual use research of concern |
| ☒ | ☐ Plants |

## Methods

| n/a | Involved in the study |
|---|---|
| ☒ | ☐ ChIP-seq |
| ☒ | ☐ Flow cytometry |
| ☒ | ☐ MRI-based neuroimaging |

## Antibodies

| | |
|---|---|
| Antibodies used | N/A |
| Validation | N/A |

# Eukaryotic cell lines

Policy information about cell lines and Sex and Gender in Research

| | |
|---|---|
| Cell line source(s) | N/A |
| Authentication | N/A |
| Mycoplasma contamination | N/A |
| Commonly misidentified lines (See ICLAC register) | N/A |

# Palaeontology and Archaeology

| | |
|---|---|
| Specimen provenance | N/A |
| Specimen deposition | N/A |
| Dating methods | N/A |

☐ Tick this box to confirm that the raw and calibrated dates are available in the paper or in Supplementary Information.

| | |
|---|---|
| Ethics oversight | N/A |

Note that full information on the approval of the study protocol must also be provided in the manuscript.

# Animals and other research organisms

Policy information about studies involving animals; ARRIVE guidelines recommended for reporting animal research, and Sex and Gender in Research

| | |
|---|---|
| Laboratory animals | N/A |
| Wild animals | N/A |
| Reporting on sex | N/A |
| Field-collected samples | N/A |
| Ethics oversight | N/A |

Note that full information on the approval of the study protocol must also be provided in the manuscript.

# Clinical data

Policy information about clinical studies
All manuscripts should comply with the ICMJE guidelines for publication of clinical research and a completed CONSORT checklist must be included with all submissions.

| | |
|---|---|
| Clinical trial registration | N/A |
| Study protocol | N/A |
| Data collection | N/A |
| Outcomes | N/A |

# Dual use research of concern

Policy information about dual use research of concern

## Hazards

Could the accidental, deliberate or reckless misuse of agents or technologies generated in the work, or the application of information presented in the manuscript, pose a threat to:

| No | Yes | |
|---|---|---|
| ☒ | ☐ | Public health |
| ☒ | ☐ | National security |
| ☒ | ☐ | Crops and/or livestock |
| ☒ | ☐ | Ecosystems |
| ☒ | ☐ | Any other significant area |

## Experiments of concern

Does the work involve any of these experiments of concern:

| No | Yes | |
|---|---|---|
| ☒ | ☐ | Demonstrate how to render a vaccine ineffective |
| ☒ | ☐ | Confer resistance to therapeutically useful antibiotics or antiviral agents |
| ☒ | ☐ | Enhance the virulence of a pathogen or render a nonpathogen virulent |
| ☒ | ☐ | Increase transmissibility of a pathogen |
| ☒ | ☐ | Alter the host range of a pathogen |
| ☒ | ☐ | Enable evasion of diagnostic/detection modalities |
| ☒ | ☐ | Enable the weaponization of a biological agent or toxin |
| ☒ | ☐ | Any other potentially harmful combination of experiments and agents |

# Plants

| | |
|---|---|
| Seed stocks | N/A |
| Novel plant genotypes | N/A |
| Authentication | N/A |

# ChIP-seq

## Data deposition

☐ Confirm that both raw and final processed data have been deposited in a public database such as GEO.

☐ Confirm that you have deposited or provided access to graph files (e.g. BED files) for the called peaks.

| | |
|---|---|
| Data access links<br>*May remain private before publication.* | N/A |
| Files in database submission | N/A |
| Genome browser session<br>(e.g. UCSC) | N/A |

## Methodology

| | |
|---|---|
| Replicates | N/A |
| Sequencing depth | N/A |
| Antibodies | N/A |
| Peak calling parameters | N/A |
| Data quality | N/A |
| Software | N/A |

# Flow Cytometry

## Plots

Confirm that:

☐ The axis labels state the marker and fluorochrome used (e.g. CD4-FITC).

☐ The axis scales are clearly visible. Include numbers along axes only for bottom left plot of group (a 'group' is an analysis of identical markers).

☐ All plots are contour plots with outliers or pseudocolor plots.

☐ A numerical value for number of cells or percentage (with statistics) is provided.

## Methodology

| | |
|---|---|
| Sample preparation | N/A |
| Instrument | N/A |
| Software | N/A |
| Cell population abundance | N/A |
| Gating strategy | N/A |

☐ Tick this box to confirm that a figure exemplifying the gating strategy is provided in the Supplementary Information.

# Magnetic resonance imaging

## Experimental design

| | |
|---|---|
| Design type | N/A |
| Design specifications | N/A |
| Behavioral performance measures | N/A |

## Acquisition

| | |
|---|---|
| Imaging type(s) | N/A |
| Field strength | N/A |
| Sequence & imaging parameters | N/A |
| Area of acquisition | N/A |

Diffusion MRI     ☐ Used     ☐ Not used

## Preprocessing

| | |
|---|---|
| Preprocessing software | N/A |
| Normalization | N/A |
| Normalization template | N/A |
| Noise and artifact removal | N/A |
| Volume censoring | N/A |

## Statistical modeling & inference

| | |
|---|---|
| Model type and settings | N/A |
| Effect(s) tested | N/A |

Specify type of analysis:     ☐ Whole brain     ☐ ROI-based     ☐ Both

| Statistic type for inference | N/A |
| --- | --- |

(See )

| Correction | N/A |
| --- | --- |

## Models & analysis

| n/a | Involved in the study |
| --- | --- |
| ☒ | Functional and/or effective connectivity |
| ☒ | Graph analysis |
| ☒ | Multivariate modeling or predictive analysis |

| Functional and/or effective connectivity | N/A |
| --- | --- |
| Graph analysis | N/A |
| Multivariate modeling and predictive analysis | N/A |

