## [Peer Review File · Nature]

Manuscript Title: Global influence of soil texture on ecosystem water limitation

Reviewer Comments & Author Rebuttals

Reviewer Reports on the Initial Version:

Referees' comments:

Referee #1 (Remarks to the Author):

This is very interesting manuscript that explores an aspect seldom considered in the analyses of the spatial differences across sites in their sensitivity to water stress, i.e., the role of soil texture. While the significance of soil texture has long been identified, only recently have we been able to have sufficient data coverage as well as the conceptual and analytical machinery to analyse this aspect in detail at the global scale. Hence, I applaud the authors for their effort which I believe to be very important and with great potential.

My current comments relate primarily with three issues, plus some additional minor points.

a) The model analyses. I was puzzled by the plots in Figure 1e and f, the simulations for which the assumption is made that (Methods) 'full transpiration corresponds to an evaporative demand of 3kPa'. I don't think I have ever seen any study, field, lab, greenhouse, where no stomatal VPD response is observed for $VPD < 3$ kPa. I am concerned that the model parameterization is substantially biased in representing realistic VPD response curves (hence under-estimating the role played by VPD and plant hydraulic signals). Some of the other model parameters given in Table S1 also raise questions in my mind. That an average plant water potential at full transpiration is only -0.5 MPa does not seem realistic for any vegetation type other than grasses or crops in the laboratory. I suspect therefore that the range of parameter values for plant hydraulic conductance are also skewed in that direction. Also, can you please clarify why soil depth is not given among the model parameters. Is the empirical calibration of effective root length taking the place of rooting depth? How realistic is this assumption relative to assuming a realistic rooting depth profile. Again, it seems to reflect an assumption about limited rooting space with the only variable being allocation to fine roots in a single fairly narrow soil layer. Soil texture, among other things, will vary with depth and I am concerned that elements related to changes in the vertical dimension are not accounted for. Can you explain in detail your strategy with regard to model parameterization and sensitivity analyses.

b) I worry about the exclusive reliance on FLUXNET data, and that the consequent inability to distinguish between plant transpiration and soil evaporation may skew the results. Soil texture will affect both processes, but the use of EF, as opposed to other estimates of T will be severely affected in some ecosystems by the significant role by soil evaporation. A better analysis of FLUXNET data with regard to which sites were chosen in relation to this process seems necessary. In addition, I do not understand why the Sapfluxnet resource was not exploited, since it provides a direct estimate of T unaffected by soil evaporation. In this respect, this paper

<https://www.sciencedirect.com/science/article/pii/S0168192322002180>

provides a useful contrast to your results. It shows a general overall greater importance of VPD over other environmental drivers but a moderating role by soil texture, which is consistent with your

results. I would feel reassured if analyses carried out on Sapfluxnet obtained similar patterns to eddy data.

c) Finally, I found some of the figures hard to read. Figure 1d is hard to understand: first, shouldn't the line with observed soil limitations meet the dashed line of no soil limitations at sand fraction=0% on the left-hand side. Also, as plotted, it is confusing relative to Fig.2d, identical variables, but now on a log scale and the dashed line crossing the straight line. The slope of the predicted line in 2d would extrapolate to impossibly negative values of Y_{crit} for %sand = 0, again making me wonder about the effects of using EF as opposed to better estimates of T. Finally, Figure 3 is hard to interpret. I suppose the VPD in hPa is to give a similar range to the theta scale. However, had the VPD been expressed in kPa (as usual) it would give much higher values of its gradient in the insets. It would be clearer if the variables had been standardized to begin with, so that standardized slopes are compared for the two drivers.

d) Finally, while I agree on the general statement that soil texture significantly mediates the responses to VPD and soil moisture as discussed by the authors in several places, there are also several examples in the text where the additional conclusion is reached that, 'soil is on average ... the hydraulic limit', 'soil hydraulic conductivity is on average the hydraulic trigger', 'soil determines the transition from energy to water limitations', etc.etc.. I don't understand what the average refers to (given the first statement), nor do the statements fit with the theory presented in Fig.1b for clay soils, nor do they seem necessary for the overall flow of the story.

Overall, this is a very interesting manuscript that, if robust enough to withstand the inevitable observations during the reviewing process, has the potential to add an interesting novel and under-appreciated dimension to the current discussions on the impacts of drought stress on vegetation productivity, growth, and vulnerability.

I have several other smaller remarks that I would be happy to provide at a subsequent stage, if required.

Referee #2 (Remarks to the Author):

Wankmüller et al. use a combination of soil-plant model simulations, field measurements, and some global projections to argue for a dominant control of soil texture in water versus energy limitation on ecosystem function. Specifically, they show this finding through testing hypotheses about whether soil texture controls soil moisture thresholds and under which conditions (clayey soils) atmospheric conditions start to have an influence on soil moisture thresholds. The finding of soil texture dependence of soil water thresholds both in volume and potential space (acknowledging artifacts in each) does indicate a strong soil texture and overall soil control on ecosystem function. My overall feeling is that these are really neat results that potentially have wide impacts. However, I had some general confusion keeping track of different hypotheses and their presentation in the figures. For example, I find the soil texture relationship with the soil water potential thresholds to be a very neat fundamental result. The methods appear to be sound overall. However, I outline several points below (mostly related to argumentation) I am struggling with. Potentially, a lot of my concerns can be addressed with rewriting/reorganizing rather than revisions to the analysis, but this is unclear.

Major Comments

1) Just outlining some of my general confusion here first. There are seemingly two hypotheses here that are being discussed simultaneously without much partitioning. The first is if soil potential threshold depends on soil texture, this means soil controls transitions from water to energy limitation (Fig. 1d; it is a great hypothesis I agree with). The second is about how there is more variability in the relationship at higher clay fractions (Fig 1e, 1f). I suppose this is a caveat on the first hypothesis that VPD starts to take over some control while still having some soil texture control. I think these ideas need to be partitioned more clearly because the discussion of both occurs simultaneously, which feels confusing. It took me a while to figure out that these were two separate points, but Figures 2 and 3 support both hypotheses to some degree and Figure 4 supports the latter one. I think these are very neat findings, but it was a lot of work for me to keep track of these different hypotheses while reading through.

2) L91-96 show a good hypothesis that makes logical sense, but what is the following hypothesis in L96-99 and Figs. 1e, 1f based on? Are these 1e, 1f figures based on a model? Why should we expect that clay soils result in more VPD control and soil threshold variability? This seems like a leap that needs to be more developed. Specifically, I agree with the idea in Fig. 1d that a strong soil control (over plant control) would result in dependence of soil water potential thresholds on soil texture. In other words, we should expect more unexplained variability in soil thresholds if conductivity in the plant traits/VPD are limiting over soil limitations. What I don't understand is L96-99 and Fig. 1e/f that suggest more clayey soils result in less soil control/more soil threshold variability.

Two more specific questions to consider here:

a) Why do clays result in more plant limitation over soil limitation? As one counter point, clay soils have the smallest soil conductivities, which would suggest more likely soil conductance (and thus texture) limitation over plant conductance. I hear the point that soil conductance can drop rapidly for drying in sandy soils. However, sandy soils have much higher conductances already compared to clayey soils that may stay above plant conductances for a large range of wetness conditions. In other words, the stated hypothesis is more about whether soil conductance is limiting over plant conductance, not necessarily the rate of change of the conductance.

b) Why should more dependence on VPD result in more soil threshold standard deviation? I think this is more about variability unexplained by soil texture that we should see, not necessarily more total uncertainty/variability of the thresholds. I think the authors are trying to isolate/attribute to control of plant traits with a model simulation (comparing green and purple lines in Fig. 2d), which I agree with and think this is a helpful attribution. However, it is unclear why observed total variability of soil moisture thresholds would suggest the same process is occurring as well.

I agree with Fig. 1d that converting soil moisture to soil water potential reduces inherent critical threshold dependence on soil texture. Therefore, if the soil water potential threshold is still dependent on soil texture, this shows a dominant role of soil water limitation. However, I don't understand the step to Figs. 1e and 1f. A lot of moving parts go into this and thus Figs. 1e and 1f are likely snapshots of selections of many variables and might change considering other variables (aerodynamic resistance, mean climate, plant properties, etc.).

In summary, I don't understand the result in Fig. 2c and L164-176 and how this supports the results

of the paper about soil control over VPD control. Potentially, the authors are correct, but these arguments are incomplete and a more convincing link on these above points can be made here.

3) There is an idea mentioned several times about soil hydraulic conductivity being limiting and the determinant of energy to water limitation (L295, L144 in Fig2). I don't think this conductivity point is well established here (Fig. 1b is a bit hard to follow what the central argument is). I suggest (a) providing more direct evidence about this conductivity point whether with a model or measurement and (b) connecting conductivity to the threshold argument more explicitly given that conductivity is a rate and the soil water threshold is a state.

4) I am surprised that sandy soils result in both drier soil moisture thresholds (Fig. 2a) and also less negative (wetter) soil water potential thresholds (Fig. 2d). This is initially a confusing result: one says sandy soils have drier thresholds and the other suggests wetter. Is this just an artifact of converting between soil water volume and potential? In other words, it appears to be more about soil properties/soil retention curves because sandy soils naturally have less negative soil water potentials (Fig. S2).

It seems a point like this needs more discussion. If there is always such a large difference in soil water potentials between different soil textures, then wouldn't there always be a soil texture dependence on the soil water potential, regardless of what the threshold is estimated to be? There is a somewhat circular argument in here. Ultimately, the fact that the soil texture dependence is shown in both soil moisture and soil water potential allays some of these concerns. However, I think either soil moisture or water potential has such a strong dependence of soil texture already that the thresholds will inherently be based on soil texture, which is a point about the soil metric itself and not about soil versus atmospheric control specifically on thresholds that this paper is suggesting.

5) Sign convention: I spent much time interpreting how high/low and positive/negative soil water potential translates to dry versus wet. I agree with the text descriptions and generally that higher sand fraction will have drier soil moisture critical thresholds. I also had some confusion related to my point #4 above. First, I recommend selecting one convention only in plots on how water potential is presented (different uses on axes across Figs 1 and 2). Second, it seems that the soil threshold for sand is low in soil moisture space and high in soil potential space, which I thought was confusing. These cases are probably correct (point #4 above and seeing Fig. S2), but need to be pointed out so that readership that are not as familiar with soil physics don't get caught up.

Specific Comments:

L139-142: I know other model parameters are kept constant to create Fig. 2b, 2d. However, are there model parameter cases where sand fraction does not alter the soil potential threshold? It would be more convincing if this is shown.

Fig. 1d: This is a good hypothesis and figure.

L177-179: I am confused about this point related to my major comment. Fig. 1c and Fig. 2a show drier soil moisture thresholds for sandy soils. However, when converting to soil water potential, this flips and shows the opposite: less negative (wetter) thresholds for sandy soils. Is this a mistake or

does the dependence invert when converting from soil moisture volume to soil water potential?

L180-184: Note that this disagrees with Fig. 1c and previous work that finds higher soil moisture thresholds in clayey soils in soil moisture space. See Fu et al. in ref 33 here. I think this is a pedotransfer function artifact that should be discussed. See major comments.

L199-201: In line with my major comment, this point needs more development about rates of change of soil conductance. Potentially there are measurements or model-based arguments to support these points.

L265-266: This point is brought up multiple times, but there is little discussion or evidence for why this is. It is challenging to see from Fig. 1b. Since it is a central explanation, it should be given more direct evidence for the reader either with measurements or a model.

L408: Related to some of my main comments, pedotransfer functions tend to be uncertain. Some uncertainty tests on how this influences Fig. 2d relationship should be made.

L446: A clarification here, the red line in Fig 3 is based on the expected threshold for the average climatic conditions across the sites?

Referee #3 (Remarks to the Author):

The authors have revealed an original finding that resolves an ongoing debate to a good extent, i.e., relative influence of VPD and soil moisture on ecosystem function. They have used public datasets for this purpose and a well-thought and comprehensive modeling framework. Overall, they found out that soil texture, especially sandy soils, strongly influences ecosystem water limitations, making them more sensitive to soil drying than VPD.

The underlying flux data is widely used in synthesis studies similar to this and is appropriate. The approach used is robust and parameterizations and model simulations are appropriate. The uncertainties are properly reported on graphics. The segmentation of findings into sections is well designed and encourages a story building.

A couple of questions that may be the basis for suggested improvements:

1. I understand that soil drying has been represented using both volumetric water content as well as soil matric potential. However, what influence do authors think variation in AWC, i.e. field capacity and permanent wilting point have on the findings? Is some of that signature encapsulated within VWC? How would that impact interpretation?

2. How about role of soil structure? Attributes like bulk density, soil organic matter can be included in the analysis, similarly to how soil texture was addressed. Since both soil texture and structure interact to determine water availability, this would be important to address.

Related work included recent advances have been cited. The writing is lucid and simple to understand for a wide audience that may lie outside the discipline of soil physics.

Meetpal S. Kukal, Ph.D.
Penn State University

Referee #4 (Remarks to the Author):

This is a very interesting paper about the global effect of soil texture on water limitation within the soil-plant-atmosphere continuum. Since plants represent unsaturated, porous media, which are typically embedded in another porous medium (the soil) and exposed to a free-flow domain (the atmosphere), the authors present exciting data about critical soil moisture thresholds for soils with different sand fractions. The manuscript provides strong evidence that plant adjustment to dehydration in coarse-textured soils is unlikely, but possible in more fine-textured soils. The role of critical soil moisture thresholds is evaluated at the ecosystem level based on soil moisture and ecosystem flux data, indicating that ecosystems with coarse-textured soils are much less sensitive to VPD than to fine-textured soils. Although the analysis of how climate change may affect terrestrial ecosystems is limited to effect of VPD on critical soil water thresholds, the authors provide clear and convincing evidence, and interpret the available data accurately.

I hope the following comments will be useful to improve the current version of the manuscript. It is confusing to see that plant organs and tissues are not accurately described in the discussion about plant conductance. In particular, K_{xylem} seems to be considered as a synonym of K_{stem} , while xylem tissue is also included in roots and leaves. In fact, ca. 95% or more of the entire hydraulic transport pathway in plants includes xylem tissue, and only a very short fraction in roots (from the root hairs to the vascular bundle) and leaves (from the minor veins to cell walls nearby the stomata) represents outer-xylem tissue.

The authors repeatedly describe the relationship between ecosystem sensitivity to VPD versus soil moisture as debated (line 217), "highly debated" (line 20, 34, 68), and even as a controversy (line 73). Clearly, the limiting hydraulic element of the soil-plant-atmosphere continuum represents an interesting and important question, and is currently largely unclear (line 334). However, any overstatement should be avoided. For instance, I would suggest to rewrite line 20 to: It follows that the ecosystem sensitivity to VPD versus soil moisture is largely shaped by soil texture, ...

Line 54-55: The most limiting hydraulic element in the soil plant continuum is said to be characterised by a sharp drop in water potential. While a "drop" in water potential between the soil and the plant seems to be likely in most cases, with roots showing a lower water potential than the soil, there may also be a difference in water potential in a reverse direction. In this case, water potentials in plant roots might be higher (less negative) than in the soil, as has been described for instance to explain the process of hydraulic lift. Also, reverse transpiration could be considered here, i.e. foliar water uptake when VPD levels become very low.

Soil spatial heterogeneity and other constraints in modelling fluxes along the soil-plant-atmosphere are said to be important challenges and limitations to the analyses conducted (e.g. line 81). Yet, it would be useful to see a more detailed discussion with some key references about the variability of the sand fraction in a soil, and especially the heterogeneous nature of soil profiles. How does variation in sand fraction, root morphology and rooting depth affect the analyses conducted? Along

the same lines, it would be interesting to provide some references or general information about morphological or anatomical plant adaptation to soil types with different sand or clay fractions. The physical constraints of different soil types are certainly important in regulating water availability to plants, but plant growth and survival is also affected by biotic interactions, such as the soil microbiome, mycorrhizae, etc. This topic would benefit from some additional discussion. Finally, I believe that all soil texture information from Eddy-Covariance sites, and the codes to analyse and visualise the data should be made publicly available if this paper will be published.

Referee #5 (Remarks to the Author):

The manuscript "Global influence of soil texture on ecosystem water limitation" analyses observations of critical soil moisture thresholds across global measurement networks. The authors show that soil texture plays a prominent role in the onset of ecosystem water limitation. The authors highlight that the controversial ecosystem sensitivity to VPD versus soil moisture is influenced by soil texture, with ecosystems in sandy soils being more sensitive to soil moisture limitation than VPD. The authors conclude that vegetation-atmosphere exchanges are driven by climatic conditions, mediated by plant adaptations, but determined by soil texture. I found the article well written, the analysis sound and it contributes to the current debate on the relative importance of atmospheric dryness versus soil water availability in determining vegetation responses. I read the manuscript with great interest and found the discussion and conclusions robust and in line with the results. The figures are clear, and the statistics well reported. The presentation of the results is clear and robust. I found that some parts of the methods could be written more clearly to allow reproducibility, and I will give some examples below.

A general comment from lines 60-68

I agree with the described mechanisms. However, these mechanisms interact in time and in my opinion it's the temporal variability of these two mechanisms that determines when the vegetation is energy limited or water limited, but it should be made clear that this can also change in time and not only the soil conditions. I think that in general the authors should really clarify at what time scale the different processes operate and when they talk about spatial variation in soil texture control, vs.

Line 96-99

I do not fully agree with this hypothesis because if the soil is well-watered, the ecosystem response to VPD should be determined by the vegetation and its stomatal response to VPD and not too much by the soil. If, instead, the authors are referring to average long-term VPD sensitivity, which is also determined by the seasonal evolution of plant water availability and VPD, then I would agree, because soil texture can determine the onset of soil moisture limitations. In other words, in two ideal ecosystems with the same precipitation, VPD and similar vegetation and leaf area index, the soil texture will determine the onset of soil water limitations and thus shape VPD sensitivity. The mechanism described by the authors depends critically on the time scale and this should be clarified.

I also have a few points to clarify in the methods section

Line 385-387

If soil texture classification was not available, we classified the soils based on the USDA soil texture classification system using the reported proportions of sand, silt and clay.

The authors should test whether the USDA soil texture classification is accurate by comparing the values for the 44 sites where in situ texture is available. As soil texture is the key element of the article, I think the authors should check whether the soil texture and USDA classification are consistent for the remaining sites.

Also, it is not entirely clear to me whether the number of sites with texture information is really 44, because in Table S3 only a few have soil texture (silt, clay and sand content). Please clarify this and I believe that much more site texture data can be compiled through the BADM and other papers.

Lines 459-460

Sufficient wind speed ($> 1 \text{ m s}^{-1}$) to promote vegetation-atmosphere coupling;

Please clarify why this threshold has been chosen. This threshold should primarily depend on vegetation height, among other factors.

It is also unclear to me whether or not gap filled data were used. In general, I found few parts of the methods unclear and could be clarified for full reproducibility.

Line 505

I believe that the codes should be published with the paper to allow full reproducibility.

Author Rebuttals to Initial Comments:

Answers to Referees “Global influence of soil texture on ecosystem water limitation”

**Referees comments in italics - Answer in blue*

Summary of changes

We thank the reviewers for their suggestions. We have modified the manuscript accordingly. The major changes are the following:

- We have added a new set of data from a different monitoring network, sapfluxnet, as suggested by ref1. The new data are consistent with the previous ones and confirm the hypothesis. This addition improves the robustness and validity of our analysis (Fig2).*
- We have repeated the simulations with a different set of plant hydraulic parameters and run a sensitivity analysis (as suggested by ref1). The additional simulations reinforce our analysis and conclusions. We have also improved the model description and the underlying assumptions, as suggested by ref5.*
- We have separated the two hypotheses: 1) the effect of texture on critical soil water thresholds (Fig1-2), and 2) the effect of soil texture on ecosystem sensitivity to VPD and soil drying (Fig3-4), as suggested by ref2. We have also tried to clarify the theoretical aspects on the relation between water potential, water content and hydraulic conductivity across different soils. A new figure (Fig1d,e) has been added to clarify this.*
- We have linked our analysis to field capacity and wilting point, as suggested by ref4 (Fig1d).*
- We have expanded the discussion on soil structure and in general on soil-plant interactions, including the effect of rhizosphere processes on water limitation, as suggested by ref4.*
- We have toned down the statements claiming the primary role of soil limitation over plant hydraulics and VPD. We acknowledge that these statements were not precise and well-balanced, whereas we highlighted that the relative importance of soil versus VPD and of soil vs plant hydraulics is soil texture specific.*

These changes, besides improving the readability of the manuscript, also improved the robustness of our analysis and the generality of our conclusions. We thank a lot the referees for many useful suggestions, which have substantially improved the manuscript.

Below, you find a point-to-point answer to the referees’ criticisms and suggestions.

Referee #1 (Remarks to the Author):

This is very interesting manuscript that explores an aspect seldom considered in the analyses of the spatial differences across sites in their sensitivity to water stress, i.e., the role of soil texture. While the significance of soil texture has long been identified, only recently have we been able to have sufficient data coverage as well as the conceptual and analytical machinery to analyse this aspect in detail at the global scale. Hence, I applaud the authors for their effort which I believe to be very important and with great potential.

My current comments relate primarily with three issues, plus some additional minor points.
a) The model analyses. I was puzzled by the plots in Figure 1e and f, the simulations for which the

assumption is made that (Methods) 'full transpiration corresponds to an evaporative demand of 3kPa'. I don't think I have ever seen any study, field, lab, greenhouse, where no stomatal VPD response is observed for $VPD < 3$ kPa. I am concerned that the model parameterization is substantially biased in representing realistic VPD response curves (hence under-estimating the role played by VPD and plant hydraulic signals). Some of the other model parameters given in Table S1 also raise questions in my mind. That an average plant water potential at full transpiration is only -0.5 MPa does not seem realistic for any vegetation type other than grasses or crops in the laboratory. I suspect therefore that the range of parameter values for plant hydraulic conductance are also skewed in that direction. Also, can you please clarify why soil depth is not given among the model parameters. Is the empirical calibration of effective root length taking the place of rooting depth? How realistic is this assumption relative to assuming a realistic rooting depth profile. Again, it seems to reflect an assumption about limited rooting space with the only variable being allocation to fine roots in a single fairly narrow soil layer. Soil texture, among other things, will vary with depth and I am concerned that elements related to changes in the vertical dimension are not accounted for. Can you explain in detail your strategy with regard to model parameterization and sensitivity analyses.

These are important points that we are happy to address.

The 3 kPa threshold of the previous Fig1 (now in Fig3) is only exemplary, and does not affect the estimation of critical soil water thresholds. However, we agree that 3 kPa for an average "full transpiration" with no stomatal closure is high. We have changed it to 1.5 kPa and expanded the VPD axis to acknowledge the effects of high VPD on stomatal conductance also under wet soil conditions. However, we would like to stress that in this study we did not investigate the sensitivity to VPD but to soil drying.

The model is sensitive to T_{max}/K_{plant} . The ratio between T_{max} and plant conductance K_{plant} gives the leaf water potential at midday in wet soils at average (unconstrained) transpiration rate. This is an important parameter to be discussed. In the former analysis we chose a value of -0.5 MPa, because we had referred to daily average transpiration and not to daily maxima. However, to test the robustness of the analysis we have now decreased the value, as suggested, to -1 MPa. The results did not change. To further investigate this point, we made a sensitivity analysis of the parameters. We found that a key factor was the difference between T_{max}/K_{plant} and ψ_{x*} , which in the model is the plant water potential at which K_{plant} decreases (similar to xylem P50). When the difference is smaller than ~ 1 MPa, soil thresholds are controlled by plants solely and ψ_{crit} does not vary with contrasting texture (clay vs sand both at around -0.05 MPa). When the difference increases above 1 MPa, then the difference between soil textures emerges. Note that in our sensitivity analysis we let the parameter T_{max}/K_{plant} change from -0.5 to -1.25 MPa and ψ_{x*} from -1.5 to -5 MPa. As the measurements show that ψ_{crit} changes with soil texture, we expect the difference between T_{max}/K_{plant} and ψ_{x*} is on average larger than ca. 1 MPa. Note that this is an average value and the large variability of ψ_{crit} in loamy soils (Fig2) indicates that there are cases in which this difference might be smaller. We added the figure below and this discussion in the supplementary material (Fig. S3).

Fig. S3: Sensitivity of ψ_{crit} on plant traits across two contrasting soil textures (clay in blue and loamy sand in green) for two root length (the reference one used for the simulations of Fig2 and one with 5 times less roots). ψ_x^* is the water potential at which plant conductance drops (e.g. due to cavitation). A key variable to explain the sensitivity of ψ_{crit} to soil texture is: $-T_{max}/K_{plant} - \psi_x^*$. When it is low ($< ca. 1$ MPa), plant limits transpiration and there are no effects of soil texture on ψ_{crit} . The effects of soil texture are visible when $-T_{max}/K_{plant} - \psi_x^* > 1$ MPa. In summary, we have increased $-T_{max}/K_{plant}$, as suggested, to -1 MPa but as long as this value is smaller than ψ_x^* , this does not substantially change the analysis.

Concerning soil depth. This is an important point. We used the soil moisture measured in the top soil layer (as in Fu et al. 2022). The justification is that the initial decline in transpiration is driven by a decline in water content in the soil layers with the highest root length density and root water uptake, which is typically the top soil layer. This is supported by the good correlation coefficient between soil moisture thresholds estimated from different soil depths (Fu et al. 2022). Uptake from deeper layers is of course important for the total transpiration and for maintaining transpiration as the soil progressively dries (i.e. the slope of T as a function of soil moisture after the critical point), but less for predicting the onset of water limitation.

In this way, our model parameter L_{root} , an effective root length for water uptake, controls the root surface area and therefore determines the water flux density (a velocity) at the root-soil interface. This velocity, required to supply the transpiration stream, determines the required soil matric potential gradients around the roots resulting in the calculation of water potentials at the root-soil interface, in the root (xylem) and in the leaves. Given that the study's focus is the onset of water limitation, the simplification to a single representative soil layer seems appropriate in our view, and it is justified by the good match with the observations.

b) I worry about the exclusive reliance on FLUXNET data, and that the consequent inability to distinguish between plant transpiration and soil evaporation may skew the results. Soil texture will affect both processes, but the use of EF, as opposed to other estimates of T will be severely affected in some ecosystems by the significant role by soil evaporation. A better analysis of FLUXNET data with regard to which sites were chosen in relation to this process seems necessary. In addition, I do not understand why the Sapfluxnet resource was not exploited, since it provides a direct estimate of T unaffected by soil evaporation. In this respect, this paper <https://www.sciencedirect.com/science/article/pii/S0168192322002180> provides a useful contrast to your results. It shows a general overall greater importance of VPD over other environmental drivers but a moderating role by soil texture, which is consistent with your results. I would feel reassured if analyses carried out on Sapfluxnet obtained similar patterns to eddy data.

We thank the reviewer for the suggestion. We have included in the analysis the Sapfluxnet dataset and found similar controlling effects of soil texture on sapflux. The new analysis fully supports our conclusion and improves the robustness and validity of our study.

Furthermore, our analysis of FLUXNET using EF refers to previous studies which proved that EF-based soil moisture thresholds correlate well with GPP-based soil moisture thresholds. This indicates that the EF-based thresholds are unlikely to be too skewed by soil evaporation as GPP scales with transpiration but not with soil evaporation.

c) Finally, I found some of the figures hard to read. Figure 1d is hard to understand: first, shouldn't the line with observed soil limitations meet the dashed line of no soil limitations at sand fraction=0% on the left-hand side. Also, as plotted, it is confusing relative to Fig.2d, identical variables, but now on a log scale and the dashed line crossing the straight line. The slope of the predicted line in 2d would extrapolate to impossibly negative values of Y_{crit} for %sand = 0, again making me wonder about the effects of using EF as opposed to better estimates of T. Finally, Figure 3 is hard to interpret. I suppose the VPD in hPa is to give a similar range to the theta scale. However, had the VPD been expressed in kPa (as usual) it would give much higher values of its gradient in the insets. It would be clearer if the variables had been standardized to begin with, so that standardized slopes are compared for the two drivers.

We have redrawn and improved the Figures accordingly. Fig.1d should be read in a conceptual way. It is intended to show a trend in $-\psi_{crit}$ with soil texture based on simulations excluding soil limitations, and on simulations including soil limitation (default). Accordingly, it does not imply that the hypothesized soil texture dependence (red line) would perfectly match the no-soil limitation at 0% sand fraction. In fact, we do not even expect that the relation of ψ_{crit} to sand fraction would be linear (why we use a locally weighted average to illustrate our hypothesis). Please see the new figures which have been modified. Also, the key variable here is the soil hydraulic conductivity curve and not the texture. We took the texture because its properties are more easily accessible and understandable. VPD is now given in kPa (as in Fig. 3a-d). The absolute values of the average relative sensitivity (inset arrows) are not relevant here, but it is rather the comparison between sand and clay sites. Please reconsider the methods for the details of the calculation of these arrows.

d) Finally, while I agree on the general statement that soil texture significantly mediates the responses to VPD and soil moisture as discussed by the authors in several places, there are also several examples in the text where the additional conclusion is reached that, 'soil is on average ... the hydraulic limit', 'soil hydraulic conductivity is on average the hydraulic trigger', 'soil determines the transition from energy to water limitations', etc.etc.. I don't understand what the average refers to (given the first statement), nor do the statements fit with the theory presented in Fig.1b for clay soils, nor do they seem necessary for the overall flow of the story.

We followed the suggestion of the reviewer and removed the statements, where we had concluded that the 'soil ... is on average ... the hydraulic limit'

Overall, this is a very interesting manuscript that, if robust enough to withstand the inevitable observations during the reviewing process, has the potential to add an interesting novel and under-appreciated dimension to the current discussions on the impacts of drought stress on vegetation productivity, growth, and vulnerability. I have several other smaller remarks that I would be happy to provide at a subsequent stage, if required.

Thank you again for the suggestions. We look forward to your other remarks.

Referee #2 (Remarks to the Author)

Wankmüller et al. use a combination of soil-plant model simulations, field measurements, and some global projections to argue for a dominant control of soil texture in water versus energy limitation on ecosystem function. Specifically, they show this finding through testing hypotheses about whether soil texture controls soil moisture thresholds and under which conditions (clayey soils) atmospheric conditions start to have an influence on soil moisture thresholds. The finding of soil texture dependence of soil water thresholds both in volume and potential space (acknowledging artifacts in each) does indicate a strong soil texture and overall soil control on ecosystem function. My overall feeling is that these are really neat results that potentially have wide impacts. However, I had some general confusion keeping track of different hypotheses and their presentation in the figures. For example, I find the soil texture relationship with the soil water potential thresholds to be a very neat fundamental result. The methods appear to be sound overall. However, I outline several points below (mostly related to argumentation) I am struggling with. Potentially, a lot of my concerns can be addressed with rewriting/reorganizing rather than revisions to the analysis, but this is unclear.

We thank the reviewer for the general appreciations and suggestions to improve the readability of the manuscript. We have separated the two hypotheses, effect of soil texture on soil water thresholds and effect of soil texture on VPD vs soil drying sensitivity.

Major Comments: 1) Just outlining some of my general confusion here first. There are seemingly two hypotheses here that are being discussed simultaneously without much partitioning. The first is if soil potential threshold depends on soil texture, this means soil controls transitions from water to energy limitation (Fig. 1d; it is a great hypothesis I agree with). The second is about how there is more variability in the relationship at higher clay fractions (Fig 1e, 1f). I suppose this is a caveat on the first hypothesis that VPD starts to take over some control while still having some soil texture control. I think these ideas need to be partitioned more clearly because the discussion of both occurs simultaneously, which feels confusing. It took me a while to figure out that these were two separate points, but Figures 2 and 3 support both hypotheses to some degree and Figure 4 supports the latter one. I think these are very neat findings, but it was a lot of work for me to keep track of these different hypotheses while reading through.

We thank the reviewer for the suggestions. We have now separated the hypotheses and removed the VPD part from Fig. 1. Indeed, all hypotheses here are the result of the same mechanism. We tried to explain it in the introduction.

2) L91-96 show a good hypothesis that makes logical sense, but what is the following hypothesis in L96-99 and Figs. 1e, 1f based on? Are these 1e, 1f figures based on a model? Why should we expect that clay soils result in more VPD control and soil threshold variability? This seems like a leap that needs to be more developed. Specifically, I agree with the idea in Fig. 1d that a strong soil control (over plant control) would result in dependence of soil water potential thresholds on soil texture. In other words, we should expect more unexplained variability in soil thresholds if conductivity in the plant traits/VPD are limiting over soil limitations. What I don't understand is L96-99 and Fig. 1e/f that suggest more clayey soils result in less soil control/more soil threshold variability.

We have removed the panels about VPD and moved them to Fig3. They are results of the model. Your statements are correct. Sandy soils are strongly constrained by soil hydraulics, which results in limited variability in soil water thresholds. In contrast clay soils are less constrained by soil and show larger variability in these thresholds. Instead of the VPD part we added a new figure showing the mechanism from a new perspective clarifying the underlying conductivity constraints.

Two more specific questions to consider here:

a) Why do clays result in more plant limitation over soil limitation? As one counter point, clay soils have the smallest soil conductivities, which would suggest more likely soil conductance (and thus texture) limitation over plant conductance. I hear the point that soil conductance can drop rapidly for drying in sandy soils. However, sandy soils have much higher conductances already compared to clayey soils that may stay above plant conductances for a large range of wetness conditions. In other words, the stated hypothesis is more about whether soil conductance is limiting over plant conductance, not necessarily the rate of change of the conductance.

We have added a new Figure (Fig 1e) to explain our argument on the role of hydraulic conductivities. It shows that when the unsaturated hydraulic conductivity at ψ_{crit} is plotted as a function of soil matric potential, the conductivity $k(\psi_{crit})$ of sandy soils (e.g. loamy sand) is lower than that of clay soils, but higher than that of loamy soils. This is due to the two fundamental hydraulic constraints determining the onset of ecosystem water limitation: a sufficient decline in soil or plant hydraulic conductance for triggering stomatal closure as a response. Due to the rapid decline (towards the roots) of soil hydraulic conductivity in sandy soils, also the overall soil hydraulic conductance at ψ_{crit} is lower in these soils (Fig 1b). We are aware that this is not trivial, but we hope that the readers can follow the argument and showing both, conductivity (Fig. 1e) and conductance (Fig. 1b) should help to clarify in this direction.

b) Why should more dependence on VPD result in more soil threshold standard deviation? I think this is more about variability unexplained by soil texture that we should see, not necessarily more total uncertainty/variability of the thresholds. I think the authors are trying to isolate/attribute to control of plant traits with a model simulation (comparing green and purple lines in Fig. 2d), which I agree with and think this is a helpful attribution. However, it is unclear why observed total variability of soil moisture thresholds would suggest the same process is occurring as well.

The effect of VPD on soil moisture threshold is similar to the effect of plant traits, for instance K_{plant} . This is why we initially grouped the discussion of soil moisture variability and sensitivity to VPD. However, we understand that this is confusing and therefore we moved the part of VPD.

Fig. 2c shows the effects of soil texture in a slightly different way, but it is not related to VPD. Firstly, it illustrates that the observed soil moisture thresholds exhibit a decreasing variability with increasing sand fraction. Secondly, it shows (again) that in sandy soils, plant hydraulic variability (ψ_{x*} , similar to ψ_{x50}) has a smaller impact on soil moisture thresholds than in clay soils. But it also shows that the model well predicts this pattern of the observed θ_{crit} -variability decreasing with sand fraction. It has also two important implications: 1) plant hydraulic adjustments (i.e. ψ_{x50}) have a limited effect in sandy soils (green line); 2) uncertainty in predicting soil hydraulic properties from texture might result in larger errors, particularly in fine textured soils (brown line), and highlights the value of good estimates of soil hydraulic properties beyond soil texture.

I agree with Fig. 1d that converting soil moisture to soil water potential reduces inherent critical threshold dependence on soil texture. Therefore, if the soil water potential threshold is still dependent

on soil texture, this shows a dominant role of soil water limitation. However, I don't understand the step to Figs. 1e and 1f. A lot of moving parts go into this and thus Figs. 1e and 1f are likely snapshots of selections of many variables and might change considering other variables (aerodynamic resistance, mean climate, plant properties, etc.).

In summary, I don't understand the result in Fig. 2c and L164-176 and how this supports the results of the paper about soil control over VPD control. Potentially, the authors are correct, but these arguments are incomplete and a more convincing link on these above points can be made here.

We have expanded the discussion of Fig. 1 and Fig.2c and we have moved the part on VPD to the next section. We hope it is easier to follow the arguments now.

3) There is an idea mentioned several times about soil hydraulic conductivity being limiting and the determinant of energy to water limitation (L295, L144 in Fig2). I don't think this conductivity point is well established here (Fig. 1b is a bit hard to follow what the central argument is). I suggest (a) providing more direct evidence about this conductivity point whether with a model or measurement and (b) connecting conductivity to the threshold argument more explicitly given that conductivity is a rate and the soil water threshold is a state.

We thank the reviewer for the suggestion. We have now included a figure with soil conductivity (Fig.1e), explaining the fundamental hydraulic constraints of a soil-plant system. We are convinced this is an important improvement to understand the argumentation about which element may limit the overall system.

4) I am surprised that sandy soils result in both drier soil moisture thresholds (Fig. 2a) and also less negative (wetter) soil water potential thresholds (Fig. 2d). This is initially a confusing result: one says sandy soils have drier thresholds and the other suggests wetter. Is this just an artifact of converting between soil water volume and potential? In other words, it appears to be more about soil properties/soil retention curves because sandy soils naturally have less negative soil water potentials (Fig. S2).

This is the effect of the different water retention curves of sandy, silty and clay soils. Indeed, sandy soils have drier thresholds (in terms of water content, i.e. thresholds appear at lower soil moisture) but these correspond to less negative matric potentials according to the observations. In principle, θ_{crit} could have decreased with sand fraction without showing a dependency to it in the soil water potential threshold space (i.e. constant ψ_{crit} with sand fraction), but very importantly, the observations based on FLUXNET and SAPFLUXNET not only show the trivial dependence of θ_{crit} to sand fraction, but also in ψ_{crit} . Both thresholds should be discussed due to the complexity of these relations. The texture specific soil hydraulic properties are very important in this discussion and indeed control the thresholds.

It seems a point like this needs more discussion. If there is always such a large difference in soil water potentials between different soil textures, then wouldn't there always be a soil texture dependence on the soil water potential, regardless of what the threshold is estimated to be? There is a somewhat circular argument in here. Ultimately, the fact that the soil texture dependence is shown in both soil moisture and soil water potential allays some of these concerns. However, I think either soil moisture or water potential has such a strong dependence of soil texture already that the thresholds will inherently be based on soil texture, which is a point about the soil metric itself and not about soil versus atmospheric control specifically on thresholds that this paper is suggesting.

The argument on VPD has been moved to the following section ("Soil texture controls the relative importance of VPD and soil moisture for ecosystem water limitation"). Regarding the threshold

dependence of soil hydraulic property: This is exactly the link we established – but we did not assume it, we proved it. In other words, we obtain this result using a soil-plant hydraulic model in which transpiration is regulated based on soil-leaf water potential dynamics.

5) *Sign convention: I spent much time interpreting how high/low and positive/negative soil water potential translates to dry versus wet. I agree with the text descriptions and generally that higher sand fraction will have drier soil moisture critical thresholds. I also had some confusion related to my point #4 above. First, I recommend selecting one convention only in plots on how water potential is presented (different uses on axes across Figs 1 and 2). Second, it seems that the soil threshold for sand is low in soil moisture space and high in soil potential space, which I thought was confusing. These cases are probably correct (point #4 above and seeing Fig. S2), but need to be pointed out so that readership that are not as familiar with soil physics don't get caught up.*

We tried to be as simple as possible and use same variables/units for plant and soil water relations. We consistently show water potentials with a negative sign, e.g. $-\psi_{\text{soil}}$, $-\psi_{\text{crit}}$, so the magnitude of the tension moves right or up from the origin of the plots. We tried to highlight the fact, that the decrease of water potential thresholds with sand fraction is something new (was not clear before), while the decrease of soil moisture threshold with sand fraction is not really eye-popping.

Specific Comments:

L139-142: I know other model parameters are kept constant to create Fig. 2b, 2d. However, are there model parameter cases where sand fraction does not alter the soil potential threshold? It would be more convincing if this is shown.

This is a very good suggestion. Indeed, when the difference between $T_{\text{max}}/K_{\text{plant}}$ (ratio between max transpiration and plant conductance) and ψ_{x^*} , which in the model is the plant water potential at which K_{plant} decreases (similar to P50) is smaller than ~ 1 MPa, soil thresholds are controlled by plants solely and ψ_{crit} does not vary with contrasting texture (clay vs sand both at around -0.05 MPa). See also our answer to rev1. We added this figure to the supplementary material and included a second 'null hypothesis' in Figure 1d (plants can become independent from soil texture effects in different ways: either by being 'too' vulnerable/limiting that they always trigger the onset of flux downregulation before the soil conductance significantly drops; or by equalizing the soil texture effects by means of extraordinary root length adjustments: $L_{\text{root inf.}}$ (~ 0.55 MPa) in Fig. 1d is approached in every soil texture if L_{root} is increased by at least $1e09$, see also Fig. S4).

Fig. 1d: This is a good hypothesis and figure.

Thank you.

L177-179: I am confused about this point related to my major comment. Fig. 1c and Fig. 2a show drier soil moisture thresholds for sandy soils. However, when converting to soil water potential, this flips and shows the opposite: less negative (wetter) thresholds for sandy soils. Is this a mistake or does the dependence invert when converting from soil moisture volume to soil water potential?

This is not a mistake but is the result of the texture specific soil water retention curves. In coarse textured soils, the water content (and the hydraulic conductivity) drops abruptly with decreasing (more negative) water potential, resulting in a higher (less negative) critical water potential ψ at lower water content θ compared to fine textured soils. In other words, coarse soils have retention curves with lower water contents at saturation, that drops at less negative water potentials.

L180-184: Note that this disagrees with Fig. 1c and previous work that finds higher soil moisture thresholds in clayey soils in soil moisture space. See Fu et al. in ref 33 here. I think this is a pedotransfer function artifact that should be discussed. See major comments.

This is again a result of the soil hydraulic properties: the soil water potential is higher, but the soil water content is lower in coarse textured soils. This is now stated explicitly in the manuscript.

L199-201: In line with my major comment, this point needs more development about rates of change of soil conductance. Potentially there are measurements or model-based arguments to support these points.

Fig 1e clarifies this point.

L265-266: This point is brought up multiple times, but there is little discussion or evidence for why this is. It is challenging to see from Fig. 1b. Since it is a central explanation, it should be given more direct evidence for the reader either with measurements or a model.

We complemented Fig.1b with Fig.1e and highlighted the difference between absolute values and rate of change with water potential, i.e. the slope or steepness, of hydraulic conductance (Fig. 1b) and soil hydraulic conductivity curves (e.g. see the different slopes of the soil hydraulic conductivity curves in Fig.1e).

L408: Related to some of my main comments, pedotransfer functions tend to be uncertain. Some uncertainty tests on how this influences Fig. 2d relationship should be made.

Uncertainties in pedotransfer functions (and in general on soil hydraulic properties) are important. We explored the sensitivity of our θ_{crit} predictions to variable soil hydraulic parameterization within a soil textural class. This result is shown by the brown line in Fig. 2c.

L446: A clarification here, the red line in Fig 3 is based on the expected threshold for the average climatic conditions across the sites?

Yes, we anchored the red line by the 99th percentile of the average (median) VPD distribution in wet soil ($\theta > \theta_{crit}$) across the 5 sites in each soil textural class.

Referee #3 (Remarks to the Author): Just minor points to comment/add

The authors have revealed an original finding that resolves an ongoing debate to a good extent, i.e., relative influence of VPD and soil moisture on ecosystem function. They have used public datasets for this purpose and a well-thought and comprehensive modeling framework. Overall, they found out that soil texture, especially sandy soils, strongly influences ecosystem water limitations, making them more sensitive to soil drying than VPD.

The underlying flux data is widely used in synthesis studies similar to this and is appropriate. The approach used is robust and parameterizations and model simulations are appropriate. The uncertainties are properly reported on graphics. The segmentation of findings into sections is well designed and encourages a story building.

A couple of questions that may be the basis for suggested improvements:

1. I understand that soil drying has been represented using both volumetric water content as well as soil matric potential. However, what influence do authors think variation in AWC, i.e. field capacity and permanent wilting point have on the findings? Is some of that signature encapsulated within VWC? How would that impact interpretation?

The reviewer correctly highlights two key points of the water retention curve. We addressed the role of field capacity and wilting point in the new figures 1d and 1e. The field capacity θ_{FC} marks the critical soil water potential in case of highly limiting root conductance. The water content at the permanent wilting point θ_{PWP} is always lower than the critical water content (see solid and dashed horizontal line in new figure 1d, note this is in water potential). The findings of this study could also be relevant to expand the definition of AWC, with the amount $\Delta z \cdot (\theta_{FC} - \theta_{crit})$ in a soil layer of thickness Δz as amount of water than can be withdrawn without reduction of transpiration rate. This amount may be more relevant compared to the standard definition of plant available water as $\Delta z \cdot (\theta_{FC} - \theta_{PWP})$.

2. How about role of soil structure? Attributes like bulk density, soil organic matter can be included in the analysis, similarly to how soil texture was addressed. Since both soil texture and structure interact to determine water availability, this would be important to address.

This is an important point which we agreed to include. Indeed, the mechanism we explained is based on soil hydraulic properties, which are in part explained by texture, but which are also affected by other variables, such as the clay mineral type, arrangement of soil organic matter and soil structure in general. These effects are particularly important in fine-textured soils, in which we observed the largest variability of soil moisture thresholds. We have added a discussion on this in the main text. However, compared to other hydrological processes of the vadose zone (water infiltration, run-off, percolation to deeper layers), the effect of soil structure on hydraulic properties is much smaller for root water uptake and evapotranspiration, because water flow in structural pores is mainly relevant for water contents close to saturation.

Related work included recent advances have been cited. The writing is lucid and simple to understand for a wide audience that may lie outside the discipline of soil physics.

Meetpal S. Kukal, Ph.D., Penn State University

Referee #4 (Remarks to the Author): comments on representation of plant biology and adaptations. Expand the discussion on vertical gradients in soils, rhizo biology, root adaptation across texture

This is a very interesting paper about the global effect of soil texture on water limitation within the soil-plant-atmosphere continuum. Since plants represent unsaturated, porous media, which are typically embedded in another porous medium (the soil) and exposed to a free-flow domain (the atmosphere), the authors present exciting data about critical soil moisture thresholds for soils with different sand fractions. The manuscript provides strong evidence that plant adjustment to dehydration in coarse-textured soils is unlikely, but possible in more fine-textured soils. The role of critical soil moisture thresholds is evaluated at the ecosystem level based on soil moisture and ecosystem flux data, indicating that ecosystems with coarse-textured soils are much less sensitive to VPD than to fine-textured soils. Although the analysis of how climate change may affect terrestrial ecosystems is limited to effect of VPD on critical soil water thresholds, the authors provide clear and convincing evidence, and interpret the available data accurately. I hope the following comments will be useful to improve the current version of the manuscript.

We thank the reviewer for the general appreciation and for the suggestions below.

It is confusing to see that plant organs and tissues are not accurately described in the discussion about plant conductance. In particular, K_{xylem} seems to be considered as a synonym of K_{stem} , while xylem tissue is also included in roots and leaves. In fact, ca. 95% or more of the entire hydraulic transport pathway in plants includes xylem tissue, and only a very short fraction in roots (from the root hairs to the vascular bundle) and leaves (from the minor veins to cell walls nearby the stomata) represents outer-xylem tissue.

We admit that our presentation and discussion of plant properties was very short, and we expanded it in the revised version. We changed K_{xylem} into K_{stem} to be consistent with the use of terms such as K_{root} , K_{leaf} .

The authors repeatedly describe the relationship between ecosystem sensitivity to VPD versus soil moisture as debated (line 217), “highly debated” (line 20, 34, 68), and even as a controversy (line 73). Clearly, the limiting hydraulic element of the soil-plant-atmosphere continuum represents an interesting and important question, and is currently largely unclear (line 334). However, any overstatement should be avoided. For instance, I would suggest to rewrite line 20 to: It follows that the ecosystem sensitivity to VPD versus soil moisture is largely shaped by soil texture, ...

We agree and we have toned down several sentences, including the suggested one.

Line 54-55: The most limiting hydraulic element in the soil plant continuum is said to be characterised by a sharp drop in water potential. While a “drop” in water potential between the soil and the plant seems to be likely in most cases, with roots showing a lower water potential than the soil, there may also be a difference in water potential in a reverse direction. In this case, water potentials in plant roots might be higher (less negative) than in the soil, as has been described for instance to explain the process of hydraulic lift. Also, reverse transpiration could be considered here, i.e. foliar water uptake when VPD levels become very low.

These are indeed important hydraulic processes, but we preferred to focus on processes most relevant at the onset of water limitations, such as high VPD and progressive soil drying, and do not mention specific circumstances where hydraulic lift and reverse flow may significantly contribute to water redistribution.

Soil spatial heterogeneity and other constraints in modelling fluxes along the soil-plant-atmosphere are said to be important challenges and limitations to the analyses conducted (e.g. line 81). Yet, it would be useful to see a more detailed discussion with some key references about the variability of the sand fraction in a soil, and especially the heterogeneous nature of soil profiles.

We thank the reviewer and agree with the suggestion. We have expanded the discussion on soil challenges, the dynamic and interacting nature of soil structure. The effect of variability in sand content is implicitly shown in Fig.2c, where we examined the effect of variability in soil hydraulic properties within each textural class (brown line, Fig.2c). This variability is the result of variability in sand content, but also in soil structure, organic matter, and bulk density. See the methods description, Table S2 and Fig. S6 for more details about the changes in soil hydraulic properties.

How does variation in sand fraction, root morphology and rooting depth affect the analyses conducted? Along the same lines, it would be interesting to provide some references or general information about morphological or anatomical plant adaptation to soil types with different sand or clay fractions.

We provide a sensitivity analysis of our results, i.e. soil texture dependence of soil water thresholds to changes in model parameters, such as sand fraction (across and within soil textural class) and active root length in the main (Fig. 2c) as well as some more details in the supplement.

We have added references on root plasticity as a function of soil texture. Schenk and Jackson (2005) showed a dependency of root depth to soil texture. Poeplau and Kätterer (2017) demonstrated that the smallest root:shoot ratio (0.10) occurred in a clay loam soil and the largest (0.22) was in a sandy soil. Vetterlein et al. (2022) observed thicker roots in sandy soils compared to loamy ones, both in column experiments and in the field. Despite the consistent observation of more roots in sandy than in clay soils, our data do not show the expected effect of such plasticity. We added this paragraph: "Plants grow more and thicker roots in sandy soils (Vetterlein et al. 2022), and have denser and longer root hairs in large (relative to root diameter) pores (White and Kirkegaard 2010), which is consistent with the greater root hair length and density under low soil moisture, i.e. in drier soils (Mackay and Barber 1985; Duddek et al. 2024). Therefore, we expected that our model predictions, which assumed identical plant traits across soils, would have overestimated θ_{crit} in coarse and underestimated it in fine textured soils. As this was not the case, in fact it was rather the opposite, it suggests the occurrence of additional limitations in coarse textured soils, such as loss of root to soil contact (Duddek et al. 2022)."

The physical constraints of different soil types are certainly important in regulating water availability to plants, but plant growth and survival is also affected by biotic interactions, such as the soil microbiome, mycorrhizae, etc. This topic would benefit from some additional discussion.

We thank the reviewer for the suggestion. We have added a discussion on rhizosphere plasticity including biotic interactions, such as the symbiosis with mycorrhiza.

Finally, I believe that all soil texture information from Eddy-Covariance sites, and the codes to analyse and visualise the data should be made publicly available if this paper will be published.

All codes and data will be made available.

Referee #5 (Remarks to the Author):

The manuscript "Global influence of soil texture on ecosystem water limitation" analyses observations of critical soil moisture thresholds across global measurement networks. The authors show that soil texture plays a prominent role in the onset of ecosystem water limitation. The authors highlight that the controversial ecosystem sensitivity to VPD versus soil moisture is influenced by soil texture, with ecosystems in sandy soils being more sensitive to soil moisture limitation than VPD. The authors conclude that vegetation-atmosphere exchanges are driven by climatic conditions, mediated by plant adaptations, but determined by soil texture. I found the article well written, the analysis sound and it contributes to the current debate on the relative importance of atmospheric dryness versus soil water availability in determining vegetation responses. I read the manuscript with great interest and found the discussion and conclusions robust and in line with the results. The figures are clear, and the statistics well reported. The presentation of the results is clear and robust. I found that some parts of the methods could be written more clearly to allow reproducibility, and I will give some examples below.

We thank the reviewer for the general interest and for the suggestions.

A general comment from lines 60-68

I agree with the described mechanisms. However, these mechanisms interact in time and in my opinion it's the temporal variability of these two mechanisms that determines when the vegetation is energy limited or water limited, but it should be made clear that this can also change in time and not only the soil conditions. I think that in general the authors should really clarify at what time scale the different processes operate and when they talk about spatial variation in soil texture control, vs.

Discussing the temporal scale of the mechanism is indeed very important and we expanded this part in the revised manuscript (Methods). The model instead is based on a diurnal regulation of transpiration (stomata open and close daily with time scale of several minutes to hours). Soil drying has the time scale of weeks. During this time, plant hydraulics changes (particularly for crops and grasses), but we assume that relevant hydraulic variables change proportionally: i.e. T_{max} , K_{plant} and L_{root} are assumed to change proportionally. Of course, this assumption should be critically discussed. We have expanded the method description to include this point. We also discussed this at the end of the section on sensitivity to VPD and soil drying.

Line 96-99

I do not fully agree with this hypothesis because if the soil is well-watered, the ecosystem response to VPD should be determined by the vegetation and its stomatal response to VPD and not too much by the soil.

This is correct and we have modified the sentence as well as Fig. 3 accordingly. However, note that, especially for high sand contents, soils are “well-watered” only in a narrow range of water potentials. We moved this sentence and clarified it. We have also extended the ranges of VPD in Fig. 3 to show the sensitivity of VPD in wet conditions. To clarify it, our hypothesis is that the relative importance of soil drying over VPD is soil texture specific: in sandy soil the sensitivity to soil drying is higher than in clay soils, and so the relative importance of soils versus VPD is higher in sandy soils.

If, instead, the authors are referring to average long-term VPD sensitivity, which is also determined by the seasonal evolution of plant water availability and VPD, then I would agree, because soil texture can determine the onset of soil moisture limitations. In other words, in two ideal ecosystems with the same precipitation, VPD and similar vegetation and leaf area index, the soil texture will determine the onset of soil water limitations and thus shape VPD sensitivity. The mechanism described by the authors depends critically on the time scale and this should be clarified.

We agree with the analysis of the reviewer. In this manuscript we have not explored the seasonal scale average VPD but rather the sensitivity to soil moisture and VPD at a given point in the VPD - soil moisture space. We tried to clarify this in the revised manuscript. The longer time average response is a very important point, and we hope that it will be further elaborated in future. For this, the evolution of plant hydraulics during the season as well as their adaptation to climate and soils should be analysed. We elaborated on it at the end of the section of sensitivity to VPD and soil moisture.

I also have a few points to clarify in the methods section

Line 385-387

If soil texture classification was not available, we classified the soils based on the USDA soil texture classification system using the reported proportions of sand, silt and clay. The authors should test whether the USDA soil texture classification is accurate by comparing the

values for the 44 sites where in situ texture is available. As soil texture is the key element of the article, I think the authors should check whether the soil texture and USDA classification are consistent for the remaining sites.

Thanks for highlighting this unclarity, all soil textures are in situ determined, either reporting the soil textural class (n = 30) or the fractions of sand, silt, and clay (n = 14). To our knowledge, only very few sites report both information at the same time. Therefore, we do not see the potential to systematically test a potential bias in soil texture classification as information is lacking. Potentially, some of the variability in soil moisture thresholds within a soil textural class might originate from uncertainty in local soil texture estimation, but we expect even more variability originating from natural variability in soil hydraulic properties within a soil textural class.

Also, it is not entirely clear to me whether the number of sites with texture information is really 44, because in Table S3 only a few have soil texture (silt, clay and sand content). Please clarify this and I believe that much more site texture data can be compiled through the BADM and other papers.

Please note that 44 is the sum of all sites reporting local soil texture information. However, 30 of these sites only report the soil textural class (e.g. sandy loam) without reporting the sand, silt, and clay fractions, while 14 of the 44 sites report local estimates of sand, silt and clay fractions without reporting the soil textural class. We aimed to be able to analyze more sites where good flux, soil moisture, and concurrent soil texture data are available, yet the number of available sites with all requirements was unfortunately rather limited across FLUXNET.

Lines 459-460

Sufficient wind speed ($> 1 \text{ m s}^{-1}$) to promote vegetation-atmosphere coupling; Please clarify why this threshold has been chosen. This threshold should primarily depend on vegetation height, among other factors.

We agree that this threshold should depend on factors such as vegetation height and atmospheric stability. We used this threshold as determined by Novick et al. (2016) as a general way to minimize atmospheric stability effects.

It is also unclear to me whether or not gap filled data were used. In general, I found few parts of the methods unclear and could be clarified for full reproducibility.

Thanks for highlighting this unclarity in the methods section. We used both measured (quality flag = 0) and good-quality gap filled (quality flag = 1) daily eddy covariance data for the main analysis, i.e. simulation of critical soil water thresholds, but for the comparison between clay and sand within the VPD-soil moisture space analysis (Fig. 3), we used as much data as possible, i.e. maximum available time resolution – (half)hourly fluxes – and all levels of gap-filled eddy covariance data. We state it now explicitly in the manuscript. Moreover, we will provide our codes to analyze the data for full reproducibility.

Line 505

I believe that the codes should be published with the paper to allow full reproducibility.

Codes and data will be available (see Manuscript).

References:

- Duddek, Patrick, Andrea Carminati, Nicolai Koebernick, Luise Ohmann, Goran Lovric, Sylvain Delzon, Celia M Rodriguez-Dominguez, Andrew King, and Mutez Ali Ahmed. 2022. 'The Impact of Drought-Induced Root and Root Hair Shrinkage on Root–Soil Contact'. *Plant Physiology*, March, kiac144. <https://doi.org/10.1093/plphys/kiac144>.
- Duddek, Patrick, Andreas Papritz, Mutez Ahmed, Goran Lovric, and Andrea Carminati. 2024. 'Observations of Root Hair Patterning in Soils: Insights from Synchrotron-Based X-Ray Computed Microtomography'. *Plant and Soil* accepted.
- Fu, Zheng, Philippe Ciais, David Makowski, Ana Bastos, Paul C. Stoy, Andreas Ibrom, Alexander Knohl, et al. 2022. 'Uncovering the Critical Soil Moisture Thresholds of Plant Water Stress for European Ecosystems'. *Global Change Biology* 28 (6): 2111–23. <https://doi.org/10.1111/gcb.16050>.
- Mackay, A. D., and S. A. Barber. 1985. 'Effect of Soil Moisture and Phosphate Level on Root Hair Growth of Corn Roots'. *Plant and Soil* 86 (3): 321–31. <https://doi.org/10.1007/BF02145453>.
- Novick, Kimberly A., Darren L. Ficklin, Paul C. Stoy, Christopher A. Williams, Gil Bohrer, A. Christopher Oishi, Shirley A. Papuga, et al. 2016. 'The Increasing Importance of Atmospheric Demand for Ecosystem Water and Carbon Fluxes'. *Nature Climate Change* 6 (11): 1023–27. <https://doi.org/10.1038/nclimate3114>.
- Poeplau, C., and T. Kätterer. 2017. 'Is Soil Texture a Major Controlling Factor of Root:Shoot Ratio in Cereals?' *European Journal of Soil Science* 68 (6): 964–70. <https://doi.org/10.1111/ejss.12466>.
- Schenk, H. Jochen, and Robert B. Jackson. 2005. 'Mapping the Global Distribution of Deep Roots in Relation to Climate and Soil Characteristics'. *Geoderma*, Deep regolith: exploring the lower reaches of soil, 126 (1): 129–40. <https://doi.org/10.1016/j.geoderma.2004.11.018>.
- Tracy, Saoirse R., Colin R. Black, Jeremy A. Roberts, and Sacha J. Mooney. 2013. 'Exploring the Interacting Effect of Soil Texture and Bulk Density on Root System Development in Tomato (*Solanum Lycopersicum* L.)'. *Environmental and Experimental Botany* 91 (July): 38–47. <https://doi.org/10.1016/j.envexpbot.2013.03.003>.
- Vetterlein, Doris, Maxime Phalempin, Eva Lippold, Steffen Schlüter, Susanne Schreiter, Mutez A. Ahmed, Andrea Carminati, et al. 2022. 'Root Hairs Matter at Field Scale for Maize Shoot Growth and Nutrient Uptake, but Root Trait Plasticity Is Primarily Triggered by Texture and Drought'. *Plant and Soil* 478 (1): 119–41. <https://doi.org/10.1007/s11104-022-05434-0>.
- White, Rosemary G., and John A. Kirkegaard. 2010. 'The Distribution and Abundance of Wheat Roots in a Dense, Structured Subsoil – Implications for Water Uptake'. *Plant, Cell & Environment* 33 (2): 133–48. <https://doi.org/10.1111/j.1365-3040.2009.02059.x>.

Reviewer Reports on the First Revision:

Referees' comments:

Referee #1 (Remarks to the Author):

The manuscript has been substantially improved relative to the version I previously reviewed. I do still have a couple of comments that need to be attended to.

The first one relates to a point I previously mentioned, which has been partially but not entirely corrected, i.e., the treatment of the relationship between VPD and stomatal conductance. I take the authors' point that the focus here is the response to soil water availability, however some of the statements in the text will still be confusing to a plant physiologist and must be corrected. Lines 421-427, 'In the green zone, transpiration is energy limited, in the brown zone it is water-limited' and later on, lines 540-545, 'higher soil moisture allows transpiration at g_{max} , but lower soil moisture induces partial stomatal closure already at $VPD < 1.5 \text{ kPa}$ '. The first statement is confusing. One can define water and energy limitations as done above, however stomata definitely close during the energy limitation phase, hence the name can be mis-leading. Equally, the second statement suggests that the beginning of stomatal closure will only occur well after 1.5 kPa under well watered conditions. I think there should be a clearer statement in the text that the model employed here does not intend to replicate the exact physiological mechanisms of stomatal physiology but that this is not impacting on the conclusions on transpiration, to avoid misunderstandings.

Line 454. The square root minimization for L_{root} . I suspect there is something wrong in the formula. It appears to be saying the following: one takes the difference between one observation and its simulation, squares it, takes the square root and THEN sums across observations. It seems to me the summation sign should go inside the square root. Please double check.

One final comment. After having read the global analysis at Figure 4, I was still unclear about whether the conclusion is that climate change will lead to ecosystems becoming more plant-limited or more soil-limited (the answer being obviously soil-texture dependent). I suspect the first scenario given the small changes in $\Delta \theta_{crit}$, however, a comment on this point would clarify what the analysis is pointing to.

Minor comments. Figure 3e and 3f are said to show that simulations of EF with VPD and θ 'agree well' with observations (both figure legend and text, line 260). Can you provide a quantitative assessment that goes beyond the purely visual analysis of the figure?

Line 333, comments on the analysis of Figure 4 applying to already vulnerable areas such as SW USA and Amazon. The rectangle actually appears to show not the Amazon per se, but rather the areas surrounding the arc of deforestation and the cerrado regions. The Amazon is obviously not a dry ecosystem (line 331).

Spelling mistakes are present at lines 485, 491, 510 (Fig.2D?), Fig. S3, Fig. S11 (no dots). Figure S11 strangely makes reference to GPP, which is never dealt with in the paper (no CO₂ effects on stomatal, no photosynthesis sub-model, etc.)

Referee #1 (Remarks on code availability):

There is a readme file and the code includes output figures that match those in the main text. However, while I reviewed the files, I did not run the code myself.

Referee #2 (Remarks to the Author):

I reread the manuscript and I think the reviewers did make the manuscript more accessible, especially with moving the VPD arguments later. I endorse the study given mainly Figure 2 showing the strong role of soil texture on both soil moisture and soil water potential. The further demonstration and implications for these relationships is great in Figures 3 and 4.

I do feel as if the explanations in Figures 1d and 1e are still challenging to follow and I did not fully follow the responses from the authors related to these points, which part of their solution was to move their VPD arguments later. I do understand the argument that plant limitations are stronger for finer soils, but am mostly wondering which overall sets of equations and or datasets helped us get there (what are the sources of the lines in Fig. 1d, 1e). Perhaps being more specific about where I am unclear: lines 90-156 explaining Figure 1 do not make it clear which panels are hypotheses and which are demonstrations with data (observations or model). Line 90 suggests there are hypotheses here. It appears that parts of figure 1 like in panels d and e that there are data from models and observations data. I know many of these details are listed in the methods, but it would help the flow and argumentation if a brief summary of data sources is worked into the lines 90-156 narrative.

I don't think my lack of understanding warrants reconsidering the manuscript's validity, because again I think Figure 2 gives evidence for the title of the paper. It might only hamper readability. But I at least want to make the authors aware of this potential shortcoming if other readers have difficulty following like I did.

I do not wish to remain anonymous,
Andrew Feldman

Minor comments:

Figure 1: Fig. 1b has a potential mistake where all the light blue (clay) lines are solid. There should be some dashed lines I think.

Figure 1: It would be helpful to state the sources of data for panels Fig. 1b to 1e.

Line 302: VPD is increasing everywhere, correct?

Figure 4a: a colorbar would be helpful

Figure 4b: how are clay and sand defined here? Would be helpful to state in the caption.

Line 351: especially where there is more clay fraction?

Referee #2 (Remarks on code availability):

Given restrictions on my government laptop, I was unable to install a new programming language by myself in time for the review (I code in python and needed admins to download on my computer). However, I read through their codes and followed their steps in general and I did not have any concerns. If this is not sufficient, please let me know and I can find a work around to download and run the scripts on a different device. Thank you.

Referee #4 (Remarks to the Author):

This revised version of Wankmüller et al. has improved considerably and has become a very solid paper about the role of soil texture on ecosystem water limitation. Obviously, some of the analyses are preliminary, but the authors discuss the constraints and limitations of their study carefully. I like the integration of the sapfluxnet data, the separation of the two main hypotheses that are tested, the sensitivity analysis, and how the data are interpreted in a careful, balanced approach. The figures are all clear and illustrate the key messages of the paper.

My congratulations to the authors for all efforts and impressive results! I am convinced that this paper on soil texture provides a new dimension to how we look at the impact of drought stress on vegetation.

Referee #5 (Remarks to the Author):

Dear authors,

the revised article improved in terms of clarity and all my suggestions were discussed and few of them implemented.

For me the current version of the manuscript is satisfactory and the inclusion of new datasets and modelling tests reinforce the robustness of the study.

Author Rebuttals to First Revision:

Dear Editor,

thank you very much for the decision about our manuscript. We have addressed the last suggestions of referees 1 and 2. Specifically, we have clarified the transpiration response to VPD and how our hydraulic model describes it (referee 1). Specifically, we have clarified that also in wet soils transpiration can be constrained even at relatively low VPD and in what conditions this happens (when plant hydraulics is limiting in relation to the transpiration demand). Additionally, we expanded the explanation of Fig 1d,e, as asked by referee 2.

Both points were relevant for improving the readability of the manuscript and we thank the two referees for these suggestions.

Below you can find our point-to-point answers to the referees'.

Referee #1 (Remarks to the Author):

*answers in blue

**Line numbers refer to the version "Track-Changes, All Markup"

The manuscript has been substantially improved relative to the version I previously reviewed. I do still have a couple of comments that need to be attended to. The first one relates to a point I previously mentioned, which has been partially but not entirely corrected, i.e., the treatment of the relationship between VPD and stomatal conductance. I take the authors' point that the focus here is the response to soil water availability, however some of the statements in the text will still be confusing to a plant physiologist and must be corrected. Lines 421-427, 'In the green zone, transpiration is energy limited, in the brown zone it is water-limited' and later on, lines 540-545, 'higher soil moisture allows transpiration at g_{smax} , but lower soil moisture induces partial stomatal closure already at $VPD < 1.5 \text{ kPa}$ '. The first statement is confusing. One can define water and energy limitations as done above, however stomata definitely close during the energy limitation phase, hence the name can be mis-leading. Equally, the second statement suggests that the beginning of stomatal closure will only occur well after 1.5 kPa under well watered conditions. I think there should be a clearer statement in the text that the model employed here does not intend to replicate the exact physiological mechanisms of stomatal physiology but that this is not impacting on the conclusions on transpiration, to avoid misunderstandings.

We thank the referee for the suggestion and for pointing us to an aspect on which we could have been more explicit. We addressed two points raised by the referee:

- 1) Our model "predicts the upmost limit of transpiration. It sets the maximum transpiration and corresponding stomatal conductance that plants could sustain under given soil water and VPD conditions. Obviously, stomatal conductance can be lower than this value, for instance limited by light and elevated CO_2 . In other words, our model provides a threshold for stomatal closure exclusively driven by hydraulic limitation, not including other factors that are crucial to predict

stomatal functioning below the SOL". So in this case we defined energy- and water-limited regimes purely for the framework of the plant hydraulic model we use. In reality other processes contribute to the formation of energy- and water-limited regimes as well. We changed the description, now in METHODS lines 43-48

- 2) We clarified that in wet soils stomata can close also at not too high VPD and explained under which conditions this can happen.

"Rising VPD (i.e., moving vertically in Fig. 3) triggers stomatal closure at a critical VPD, which is set by the stress onset limit (yellow line in Fig. S2). The critical VPD declines with decreasing soil water content, but it remains relatively constant for $\theta > \theta_{crit}$ (particularly in sandy soils; in Fig. 3 this critical VPD is ca. 2 kPa). This critical VPD in wet soil depends on plant hydraulics and T_{pot} . Precisely, critical VPD depends on the difference between T_{pot}/K_{plant} and the critical leaf water potential where K_{plant} declines (ψ_{star} , Supp. info). For instance, transpiration would become water-limited at a low VPD (and high soil moisture) when the plant hydraulic conductance is too low to sustain high transpiration fluxes. In this case, the system becomes water limited even in wet soils due to plant limitations and the key driver is the rising VPD.". METHODS Lines 167-181.

Line 454. The square root minimization for Lroot. I suspect there is something wrong in the formula. It appears to be saying the following: one takes the difference between one observation and its simulation, squares it, takes the square root and THEN sums across observations. It seems to me the summation sign should go inside the square root. Please double check.

The formula was not wrong, it is not the *root mean standard deviation*, but we vary L in order to find the *least absolute deviations*. However, to avoid misinterpretation, we changed to formula to an equal expression using the absolute instead of the root of squared values.

One final comment. After having read the global analysis at Figure 4, I was still unclear about whether the conclusion is that climate change will lead to ecosystems becoming more plant-limited or more soil-limited (the answer being obviously soil-texture dependent). I suspect the first scenario given the small changes in delta θ_{crit} , however, a comment on this point would clarify what the analysis is pointing to.

We agree that this is an interesting question whether climate change not only causes shifts in water limitation (e.g. regions become more frequent water limited than before), but might also shift the hydraulic limits, i.e. ecosystems being likely more plant or soil limited. Nonetheless, we believe that our simplistic climate change scenario is not well suited to overall conclude on this. We hope this will be a matter of future research.

Minor comments. Figure 3e and 3f are said to show that simulations of EF with VPD and θ 'agree well' with observations (both figure legend and text, line 260). Can you provide a quantitative assessment that goes beyond the purely visual analysis of the figure?

We agree that this part of our statement is based on the visual agreement between strong color changes (eddy fluxes) and the red line (simulations for an average soil texture). In this specific aspect, we don't provide additional analysis because we aim to highlight particularly the differences between soil textures (comparably stronger control by soil drying in sandy than in clay soils) that fit to our model predictions.

Line 333, comments on the analysis of Figure 4 applying to already vulnerable areas such as SW USA and Amazon. The rectangle actually appears to show not the Amazon per se, but rather the areas surrounding the arc of deforestation and the cerrado regions. The Amazon is obviously not a dry ecosystem (line 331).

Reviewer 1 made a good point in specifying the ecoregions around the Amazon. Indeed it's rather the cerrado (Savanna grass, low trees, shrubs), caatinga (thorny scrub) and tropical semideciduous region south of the tropical Amazon rain forest and only a small part of the southern Amazon. We refined our paragraph to "Therefore, ecosystems in coarse textured soils (highlighted by red rectangles) subject to regular water limitation may experience a high risk of exacerbating water stress and drought mortality. Strikingly, this includes some land-climate systems that are expected already or in the future being vulnerable to drought, such as the American Southwest and the southern Amazonia".

Spelling mistakes are present at lines 485, 491, 510 (Fig.2D?), Fig. S3, Fig. S11 (no dots).

Thank you for pointing out these mistakes. We have corrected them accordingly.

Figure S11 strangely makes reference to GPP, which is never dealt with in the paper (no CO2 effects on stomatal, no photosynthesis sub-model, etc.)

We agree that our simplistic model approach does not explicitly include other constraints to photosynthesis (e.g. CO₂, carboxylation capacity etc.) and transpiration (e.g. change in water-use-efficiency) other than the hydraulic soil-plant limits (see Lines 341ff.). It is indeed a promising direction to link the hydraulic limits to these other constraints. Nonetheless, we believe that Fig.S11 helps interpreting, in a general manner, the different consequences that ΔT_{pot} may have depending on soil texture and moisture regime. Note that, we touch GPP also in the main text (e.g. Lines 296 or 330).

Referee #1 (Remarks on code availability):

There is a readme file and the code includes output figures that match those in the main text. However, while I reviewed the files, I did not run the code myself.

Referee #2 (Remarks to the Author):

I reread the manuscript and I think the reviewers did make the manuscript more accessible, especially with moving the VPD arguments later. I endorse the study given mainly Figure 2 showing the strong role of soil texture on both soil moisture and soil water potential. The further demonstration and implications for these relationships is great in Figures 3 and 4.

I do feel as if the explanations in Figures 1d and 1e are still challenging to follow and I did not fully follow the responses from the authors related to these points, which part of their solution was to move their VPD arguments later. I do understand the argument that plant limitations are stronger for finer soils, but am mostly wondering which overall sets of equations and or datasets helped us get there (what are the sources of the lines in Fig. 1d, 1e). Perhaps being more specific about where I am unclear: lines 90-156 explaining Figure 1 do not make it clear which panels are hypotheses and which are demonstrations with data (observations or model). Line 90 suggests there are hypotheses here. It appears that parts of figure 1 like in panels d and e that there are data from models and observations data. I know many of these details are listed in the methods, but it would help the flow and

argumentation if a brief summary of data sources is worked into the lines 90-156 narrative.

We thank the reviewer for pointing to the challenges related to understanding Fig.1d,e. We have added some lines to explain how these curves have been derived. We made it clear that these lines and points are the results of a model, and that these are actually the hypotheses that we aimed to address. Lines 95-103, 120-123, 161-164

I don't think my lack of understanding warrants reconsidering the manuscript's validity, because again I think Figure 2 gives evidence for the title of the paper. It might only hamper readability. But I at least want to make the authors aware of this potential shortcoming if other readers have difficulty following like I did.

I do not wish to remain anonymous, Andrew Feldman

Minor comments:

Figure 1: Fig. 1b has a potential mistake where all the light blue (clay) lines are solid. There should be some dashed lines I think.

We have verified Fig.1b, all light blue (clay) lines are present, i.e. one solid (Kplant), one dashed (Ksoil), and one dashed-dotted.

Figure 1: It would be helpful to state the sources of data for panels Fig. 1b to 1e.

See answer to main comment above: "We made it clear that these lines and points are the results of a model".

Line 302: VPD is increasing everywhere, correct?

It's true for most of the world, but some regions are projected to experience a decrease in potential transpiration rate. We changed the potentially misleading "given increasing VPD" into "due to increasing VPD".

Figure 4a: a colorbar would be helpful

We agree that the colours might be difficult to disentangle, however, we cannot simplify panel b (radial coordinates) into a linear colourbar.

Figure 4b: how are clay and sand defined here? Would be helpful to state in the caption.

Thanks for pointing out the the explanation about this has come up short. We added [colors change continuously from sand (red) to clay (yellow)] "according to the soil texture specific relation between ΔT_{pot} and $\Delta \theta_{crit}$ (the colours stem from the 12 different soil textural classes; compare also the different slopes, e.g. clay vs. sand, to Fig.1c and Fig.S13)."

Line 351: especially where there is more clay fraction?

It depends: it's true that soil water thresholds can change the most in clay soil (potential), but in our analysis we find loamy soils (orange colour in Fig.4) to have highest $\Delta \theta_{crit}$ simply because some of these soils are in regions with the highest increase in T_{pot} (up to +65 %). Nonetheless, we expect ecosystems on coarse soils (red colour) to be in the most problematic situation as the water saving

effect of 'earlier' stomatal closure is strongly diminished and likely insufficient to prevent severe vegetation water stress given the acceleration of the water cycle (here: soil drying) under increasing VPD.

Referee #2 (Remarks on code availability):

Given restrictions on my government laptop, I was unable to install a new programming language by myself in time for the review (I code in python and needed admins to download on my computer). However, I read through their codes and followed their steps in general and I did not have any concerns. If this is not sufficient, please let me know and I can find a work around to download and run the scripts on a different device. Thank you.

Referee #4 (Remarks to the Author):

This revised version of Wankmüller et al. has improved considerably and has become a very solid paper about the role of soil texture on ecosystem water limitation. Obviously, some of the analyses are preliminary, but the authors discuss the constraints and limitations of their study carefully. I like the integration of the sapfluxnet data, the separation of the two main hypotheses that are tested, the sensitivity analysis, and how the data are interpreted in a careful, balanced approach. The figures are all clear and illustrate the key messages of the paper.

My congratulations to the authors for all efforts and impressive results! I am convinced that this paper on soil texture provides a new dimension to how we look at the impact of drought stress on vegetation.

Referee #5 (Remarks to the Author):

Dear authors,
the revised article improved in terms of clarity and all my suggestions were discussed and few of them implemented.
For me the current version of the manuscript is satisfactory and the inclusion of new datasets and modelling tests reinforce the robustness of the study.